# TIME TRANSFER: ON OPTIMAL LEARNING RATE AND BATCH SIZE IN THE INFINITE DATA LIMIT

## ABSTRACT

One of the main challenges in optimal scaling of large language models (LLMs) is the prohibitive cost of hyperparameter tuning, particularly learning rate $\eta$ and batch size $B$. While techniques like $\mu$P (Yang et al., 2022) provide scaling rules for optimal $\eta$ transfer in the infinite model size limit, the optimal scaling behavior in the infinite data size limit remains unknown. We fill in this gap by observing for the first time an intricate dependence of optimal $\eta$ scaling on the pretraining token budget $T$, $B$ and its relation to the critical batch size $B_{\mathrm{crit}}$, which we measure to evolve as $B_{\mathrm{crit}} \propto T$. Furthermore, we show that the optimal batch size is positively correlated with $B_{\mathrm{crit}}$: keeping it fixed becomes suboptimal over time even if learning rate is scaled optimally. Surprisingly, our results demonstrate that the observed optimal $\eta$ and $B$ dynamics are preserved with $\mu$P model scaling, challenging the conventional view of $B_{\mathrm{crit}}$ dependence solely on loss value. Complementing optimality, we examine the sensitivity of loss to changes in learning rate, where we find the sensitivity to decrease with increase of $T$ and to remain constant with $\mu$P model scaling. We hope our results make the first step towards a unified picture of the joint optimal data and model scaling.

## 1 INTRODUCTION

Large Language Models (LLMs) have increasingly become a prominent area of study in the field of Natural Language Processing (NLP) and beyond. They have demonstrated significant improvement in performance across a wide range of tasks, such as language understanding, text generation, translation, and summarization, showing results comparable or outperforming those of an average domain expert (Dubey et al., 2024; OpenAI et al., 2024; Team et al., 2024). The primary advantage of LLMs is their ability to scale well with increased computational resources, which results in predictive improved performance (Kaplan et al., 2020; Hoffmann et al., 2022).

One of the main challenges in LLM scaling lies in the proportional scaling of computational resources required for hyperparameter tuning. To remedy this, $\mu$Transfer (Yang et al., 2022) technique was proposed as a way to transfer hyperparameters from a small (proxy) model to a large (target) one by introducing scaling rules for learning rate, weight multipliers and initialization scale, altogether referred to as Maximal Update Parametrization ($\mu$P). While significantly reducing the hyperparameter tuning cost coming with model scaling, its applicability is limited by requiring both target and proxy models to share the same batch size and number of training iterations. With current pretraining budgets surpassing trillions of tokens, it makes $\mu$Transfer computationally expensive to apply even with tuning a small proxy model.

One solution would be hyperparameter tuning performed both for the small proxy model *and* on the small dataset, followed by $\mu$Transfer to the larger model and larger dataset, under assumption of both datasets being sampled from the same underlying data distribution. This raises the question of $\mu$Transfer's applicability in the *infinite data limit*, which can be formalized as an increase in the size of the training dataset, which in the LLM case is measured by the number of tokens. Understanding training dynamics in this limit would unlock hyperparameter transfer not only across model scales, but also across data horizons, thus removing the largest limitation of $\mu$Transfer.

The study of optimal hyperparameter evolution throughout the model training should also be complemented with a study of hyperparameter *sensitivity*, i.e. the measure of how the model performance is affected when the training is performed outside the optimal hyperparameter range. In practice, it

is rarely possible to remain within the optimum due to statistical uncertainties in its estimation. It would be of large interest to find training regimes which have small hyperparameter sensitivity and penalize model performance the least if the optimal hyperparameters are missed by a small degree.

Expanding on this line of research, we consider a commonly used LLM pretraining setup and aim towards building a yet missing holistic picture of optimal learning rate and batch size dynamics as one scales up the model training – both in the data and model sizes. Our main contributions are summarized as follows:

- **Optimal learning rate scaling:** by incorporating a dependence on the pretraining token budget into the theoretical model of optimal learning rate $\eta^*$ scaling Li et al. (2024) via Eq. 3.1 and performing a fit to experimentally observed data (Fig. 1), we establish a dependence of the $\eta^*$ evolution with $T$ on the batch size $B$ and its relation to the critical batch size $B_{\mathrm{crit}}$ (see definition in Sec. 2.1). From interpreting the model fit results, we obtain scaling behaviors ranging from $\eta^* \propto \sqrt{T}$ to $\eta^* \propto 1/\sqrt{T}$ depending on $B$, $B_{\mathrm{crit}}$ and $T$, which we find compatible with experimental observations. Furthermore, we find these dynamics to be largely preserved within $\mu$P (Appendix A.11).

- **Optimal batch size scaling:** assuming $\eta$ is optimal for a given data horizon $T$, we observe a gradual increase of the optimal batch size $B^*$ with an increase of the token budget (Fig. 3a). The drift is correlated with the evolution of the critical batch size $B_{\mathrm{crit}}$ (Fig. 2, left), with $B^*(T) < B_{\mathrm{crit}}(T)$ in our measured range of $T$. Importantly, we show that naïve application of optimal $\eta$ scaling rules in the $T \to \infty$ limit with $B$ being indefinitely fixed becomes suboptimal over time: a joint $(\eta, B)$ scaling is required.

- **Critical batch size:** we experimentally find $B_{\mathrm{crit}}$ (see definition in Sec. 2.1) to evolve in time with $B_{\mathrm{crit}} \propto T^{\alpha_B}$ and $\alpha_B = 1.0 \pm 0.2$ (Fig. 2, left). This dynamic affects optimal $\eta$ scaling via Eq. 3.1 and drives the transition between various scaling behaviors (Sec. 3.2). Surprisingly, we show evidence that $B_{\mathrm{crit}}$ is not exclusively defined by the value of the loss function (Eq. 8) as suggested by McCandlish et al. (2018): models within $\mu$P share the same $B_{\mathrm{crit}}$ region while having different performance in terms of loss.

- **Learning rate sensitivity:** the sensitivity is generally decreasing with an increase of the training token budget, which is interestingly more pronounced for the batch sizes in the critical batch size region (Fig. 4). We observe no significant change in the learning rate sensitivity with the change of the $\mu$P base model and within the $\mu$P width limit (Fig. 5).

## 2 METHODOLOGY

### 2.1 TERMINOLOGY

**Time ($T$):** we often use the terms *time*, *token budget*, and *data horizon* interchangeably, both to specify the measure of the training data size in tokens, and to pinpoint the specific moment throughout the model training. From this perspective, an *infinite data limit* $T \to \infty$, as opposed to a fixed budget regime with $T = \mathrm{const}$, refers to an (infinite) increase of the number of tokens seen by the model during pretraining.

**$\mu$P:** we refer to a model with width $d_{\mathrm{model}}^{\mathrm{base}}$ as a *base model* if $\mu$P scaling multipliers for learning rates, weight multipliers and initialization scale (Sec. 2.2) are computed relative to this width. This brings us to a broader view on $\mu$P where the base model "pinpoints" the training dynamics for all the other models obtained either by scaling up or down the base $d_{\mathrm{model}}^{\mathrm{base}}$ width. Together with the base model, we refer to this set of models as a *$\mu$P model family* or as a *$\mu$P trajectory* if the direction of scaling is implied. We also slightly distinguish between the base and proxy models, where the former is used to define a $\mu$P model family, while the latter is a model used to tune hyperparameters to be transferred with $\mu$Transfer to a target model.

**Critical batch size ($B_{\mathrm{crit}}$):** following Li et al. (2024), we define $B_{\mathrm{crit}}$ as the corresponding parameter in Eq. 3.1, also describing the peak position of the bell-shaped curve (Fig. 1a), which was shown by the authors to equal the $B_{\mathrm{crit}}$ definition of McCandlish et al. (2018). To better clarify the nomenclature appearing in the literature, we provide an extended discussion in Appendix A.2.

**Sensitivity:** as acknowledged by Wortsman et al. (2023), it is difficult to formalize this notion, also in the absence of a theory to be verified. We therefore define it in the most minimal way, namely as the variation of validation loss $\mathcal{L}_{\mathrm{val}}(\eta) - \mathcal{L}_{\mathrm{val}}(\eta^*)$ for a given learning rate variation from its optimal value $\eta/\eta^*$. We refer to the corresponding loss vs. learning rate curve (both with and without $\mathcal{L}_{\mathrm{val}}(\eta^*)$ normalization) as a *loss profile*.

## 2.2 MODEL CONFIGURATION AND DATASETS

For all our experiments we use a default MPT model architecture (MosaicML, 2023) as implemented in the `llm-foundry` codebase (MosaicML, 2024), with all the models sharing the same training configuration (Appendix A.3). We use the Decoupled AdamW optimizer (Loshchilov & Hutter, 2019) with $\beta_1 = 0.9$, $\beta_2 = 0.95$, $\epsilon = 10^{-8}$, weight decay $\lambda = 0$ and gradient clipping by the $L_2$ norm value of 1.

$\mu$P is implemented according to Table 8 of Yang et al. (2022), so that when $d_{\mathrm{model}}$ is set to the base model width $d_{\mathrm{model}}^{\mathrm{base}}$, it replicates Standard Parametrization (SP). That makes our observations for the base models also applicable to setups that use SP rather than $\mu$P. Model weights are initialized from the normal distribution with the base model standard deviation $\sigma^{\mathrm{base}} = 1/\sqrt{d_{\mathrm{model}}^{\mathrm{base}}}$. The models are scaled up/down only in width, with the head dimension $d_{\mathrm{head}}$ being always fixed and the number of heads being scaled proportionally to the width scaling.

The models are trained with the causal language modeling task on the train split of the Colossal Clean Crawled Corpus (C4) dataset (Raffel et al., 2020), tokenized with the GPT2 tokenizer (Radford et al., 2019) with a vocabulary size of 50257 and a context length of 1024 tokens. As a metric to evaluate model performance, we report the loss on the C4 validation split as $\mathcal{L}_{\mathrm{val}}$.

## 2.3 HYPERPARAMETER GRID

To investigate the interplay of learning rate and batch size in the infinite data limit $T \to \infty$, we define a 5D grid spanned by the following axes: $\eta$, $B$, $T$, $d_{\mathrm{model}}$, $d_{\mathrm{model}}^{\mathrm{base}}$ (see Appendix A.4 for exact definition). Fundamentally, we are interested in measuring how the loss profile $\mathcal{L}_{\mathrm{val}}(\eta)$ and its optimum value $\eta^*$ evolve in time $T$ depending on the choice of batch size $B$. As this measurement is moreover conditioned on the $\mu$P trajectory and a specific point therein, we firstly study this evolution for a trajectory pinpointed by one specific base model with $d_{\mathrm{model}}^{\mathrm{base}}$. We train a set of models within the defined $\mu$P trajectory with different widths $d_{\mathrm{model}}$, ranging in size from 32M up to 354M parameters, and measure for each of them the $\mathcal{L}_{\mathrm{val}}(\eta)$ profile at specific points in time $T$, ranging from 1B up to 275B tokens. Then, we repeat the same measurement for a new $\mu$P trajectory, pinpointed by a different value of $d_{\mathrm{model}}^{\mathrm{base}}$. This grid approach allows us to interpret results from multiple perspectives, as we detail in Sec. 3.

## 2.4 LEARNING RATE SCHEDULE SCALING

Since we study the training dynamics in the infinite data limit, it necessarily implies training models across different data horizons. This raises the question of how one should adjust the learning rate schedule in this limit. Motivated by recent work of Hu et al. (2024); Hägele et al. (2024), in all our experiments we use a warmup-stable (WS) version of the warmup-stable-decay (WSD) schedule consisting of a warmup phase with a linear increase of learning rate from 0 to $\eta_{\mathrm{max}}$ and a constant phase with learning rate fixed at $\eta_{\mathrm{max}}$, hereafter notated as $\eta$. Our version omits the decay phase to simplify experimentation as we observe that it does not affect the optimal $\eta$ position (Appendix A.7). The warmup duration is fixed across all horizons and across all experiments at an absolute value of $T_{\mathrm{warmup}} = 2^{19} = 524288$ tokens. Whenever batch size is varied, we adjust the number of gradient steps in the warmup phase accordingly so that the total amount of tokens seen by the model during warmup equals $2^{19}$. We also present additional experiments with different ways to scale the learning rate warmup and an added decay phase in Appendix A.7, with results largely confirming those of Hägele et al. (2024). The WS schedule allows us to reduce computational requirements by approximately a factor of two: contrary to retraining for each of the data horizons in the $T$ grid, we run indefinitely continued trainings and take evaluation snapshots on the way.

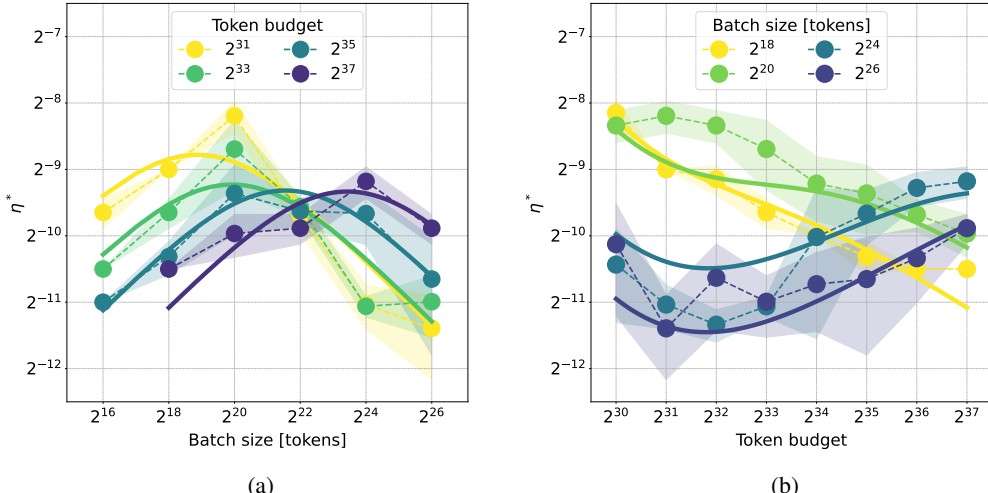

(a)  (b)

Figure 1: **(a)**: Optimal learning rate $\eta^*$ per batch size against a set of pretraining token budgets (see Appendix A.9 for a full set). Each point is obtained by averaging experimental observations of optimal learning rate values across $\mu$P model family and random seeds, as described in Sec. 3.1, with color bands visualizing the corresponding standard deviation. Solid lines represent the fitted theoretical model of Li et al. (2024) (Eq. 3.1) as described in Sec. 3.1, dashed lines only connect the data points for visualization purposes. We observe an approximately linear growth of $B_{\mathrm{crit}}$ (see also a dedicated Fig. 2), defined as the peak position of the fitted curve, in the limit of increased token budget.
**(b)** Transposition of Fig. 1a: evolution of the optimal learning rate with an increase of the pretraining token budget $\eta^*(T)$ for a representative set of batch sizes, in tokens. We observe the fitted model to describe the scaling behavior of low ($B = 2^{18}$) and high ($B = 2^{26}$) batch sizes, as well as intermediate batch sizes in the high token budget regime. For $B = 2^{18}$, the model reduces to $\eta^* \propto 1/\sqrt{T}$ as discussed in Sec. 3.2, matching the observations.

## 3 RESULTS

### 3.1 CRITICAL BATCH SIZE EVOLVES IN TIME, BUT IS UNCHANGED WITHIN $\mu$P

First, we begin with setting $d_{\mathrm{model}}^{\mathrm{base}} = 1024$ and scanning learning rate across different batch sizes and $d_{\mathrm{model}}$. We present results for the $\eta^*$ optimum dependence on the batch size $B$ per data horizon $T$ in Fig. 1a, with individual $\mathcal{L}_{\mathrm{val}}(\eta)$ profile scans in Appendix A.8 and the full set of horizons in Appendix A.9. In order to reduce statistical uncertainties, we average results across three $\mu$P models[1] with $d_{\mathrm{model}} \in \{256, 512, 1024\}$ for tokens budgets $T \leq 2^{35}$ and additionally across four more random seeds for large batch size values $B \in \{2^{20}, 2^{22}, 2^{24}, 2^{26}\}$ for the model with $d_{\mathrm{model}} = 256$ to reduce statistical fluctuations in the low token budget region. We include a similar plot for the other base model with $d_{\mathrm{model}}^{\mathrm{base}} = 256$ in Appendix A.10 and individual plots for each of the $(d_{\mathrm{model}}, d_{\mathrm{model}}^{\mathrm{base}})$ configurations in Appendix A.11.

We observe that for a given time horizon, the $(\eta^*, B)$ curve has a bell-like shape, as predicted by Li et al. (2024). The left-hand side of the peak represents a known $\eta \propto \sqrt{B}$ scaling rule (Malladi et al., 2023; Shen et al., 2024). However, with our experiments, we uncover a previously unseen right-hand side of the curve, also referred to as "surge" by Li et al. (2024), where the optimal learning rate for a fixed token budget scales inversely proportionally to the batch size scaling via the $\eta^* \propto 1/\sqrt{B}$ rule.

---

[1]We believe this averaging approach is justified since all the three models share the same optimization trajectory in terms of the number of steps, batch size and data horizon length, therefore are theoretically guaranteed by $\mu$Transfer to share the same optimal learning rate. From the experimental side, we also observe no significant differences across the three models (Appendix A.11).

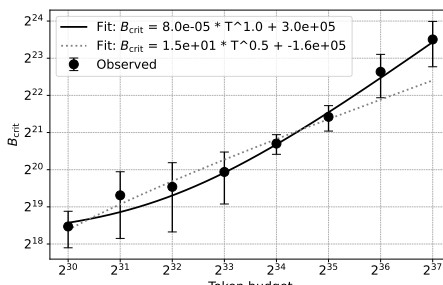 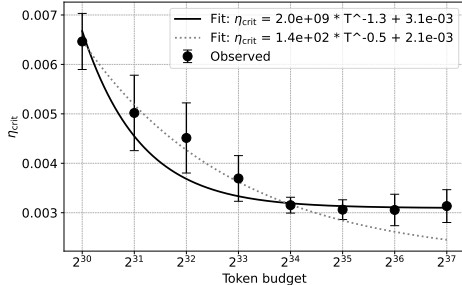

Figure 2: Critical batch size $B_{\text{crit}}$ (left) and critical learning rate $\eta_{\text{crit}}$ (right), as extracted from the fit with the power law $p_{\text{crit}} = a_p T^{\alpha_p} + b_p$, where $p \in \{\eta, B\}$, following the procedure from Appendix A.5, as a function of token budget. Solid line represents the fit result. Dashed line corresponds to the fit with the power exponent fixed to $\alpha_B = 0.5$ (left) and $\alpha_\eta = -0.5$ (right). This model fit is visualized only to illustrate the model variation with the exponent change and its parameters are not used in the main analysis.

We analyze the observed data points within the theoretical framework of Li et al. (2024). For each of the token budgets we fit the data with the following functional form:

$$\eta^*(T, B) = \frac{\eta_{\text{crit}}(T)}{\sqrt{\frac{B}{B_{\text{crit}}(T)}} + \sqrt{\frac{B_{\text{crit}}(T)}{B}}}, \tag{1}$$

where $B_{\text{crit}}$ and $\eta_{\text{crit}}$ are parameters of the fit. The former corresponds to the peak position of the bell-like curve and was shown in McCandlish et al. (2018); Li et al. (2024) to approximate the critical batch size defined as the balance point between optimal number of training steps and data efficiency. The latter can be interpreted as the optimal learning rate when training in the regime with the batch size tracking the critical one, i.e. $B(T) = B_{\text{crit}}(T)$.

After performing the fit for each token budget, we analyze the $\eta_{\text{crit}}$ and $B_{\text{crit}}$ evolution in time. We fit a power law of the form $p_{\text{crit}} = a_p T^{\alpha_p} + b_p$, $p \in \{\eta, B\}$ to each of the data sets following the procedure described in Appendix A.5 and present the results in Fig. 2 (solid lines). For the time dependence of critical parameters, we obtain the following power exponents:

$$\begin{aligned} B_{\text{crit}} &\propto T^{\alpha_B}, & \alpha_B &= 1.0 \pm 0.2, \\ \eta_{\text{crit}} &\propto T^{\alpha_\eta}, & \alpha_\eta &= -1.3 \pm 0.4, \end{aligned} \tag{2}$$

where we find that in both cases the hypothesis of the power exponent $+1$ $(-1)$ for $\alpha_B$ $(\alpha_\eta)$ is compatible with experimental observations within uncertainties. Detailed results for the $a, b$ coefficients are provided in Appendix A.5.

Lastly, there is a difference of $B_{\text{crit}}$ evolution between the $T$ and $\mu$P infinite width limits. Specifically, for a fixed token budget, we observe no significant change of the curves' shapes and peak positions across $d_{\text{model}}$ values within the same $\mu$P trajectory, and also with the change of the base model (Appendix A.11). At the same time, there is a noticeable drift of $B_{\text{crit}}$ in the $T \to \infty$ limit with the model being fixed. As both limits are accompanied with a comparable change of the model performance[2], this observation brings evidence that dependence of the critical batch size exclusively on the loss value suggested by Kaplan et al. (2020) (Eq. 8) is not entirely complete. Or, contrary to results in Li et al. (2024), the two definitions of the critical batch size region (Appendix A.2) are not the same and should be disentangled.

---

[2]Back-of-the-envelope calculation from Fig. 3a and Appendix A.12: for $d_{\text{model}}^{\text{base}} = 1024$, $B = 2^{20}$, there is a loss change $\mathcal{L}_{\text{val}} = 3.4 \to 2.8$ with a token budget increase $2^{31} \to 2^{37}$, resulting in $B_{\text{crit}}$ drifting by $2^4$. For the same $(d_{\text{model}}^{\text{base}}, B)$ configuration, there is no significant $B_{\text{crit}}$ drift with a change of width by $2^2$ within $\mu$P, but the corresponding loss change is $\mathcal{L}_{\text{val}} = 3.5 \to 2.9$.

## 3.2 LEARNING RATE OPTIMUM DRIFTS IN TIME, WITH BATCH SIZE INTERPOLATING BETWEEN DIFFERENT SCALING RULES

In Fig. 1b, we reinterpret Fig. 1a by transposing the batch size and token budget axes and by plotting the evolution of the optimal learning rate $\eta^*$ in time $T$ for a representative set of batch size values, with the full set of batch size values in Appendix A.9. Overlayed, we also plot the model fitted with Eq. 3.1 (solid lines).

From the data points alone we observe an intricate drift of the optimal learning rate in time as governed by the batch size value. In a simplified way, for small $B$ values ($2^{18}$ and $2^{20}$ in Fig. 1b), we observe a decrease of $\eta^*$ by $2^2$ with an increase of the token budget by $2^7$, while for the larger $B$ values ($2^{24}$ and $2^{26}$), it is oppositely an increase by up to $2^2$.

We also find that the fitted model[3] describes the data points for the smallest ($2^{16}$ and $2^{18}$) and largest ($2^{26}$) probed batch sizes well. For the intermediate batch size values, it captures the behavior in the large token budget regime and the general curvature patterns ($B = 2^{24}$), but sometimes lacks the correct amplitude. We note, however, large uncertainties on the fitted model parameters (Appendix A.5): additional data points with improved resolution in both $\eta$ and $B$ would better constrain the model fit and therefore constitute an important next step in future work.

It is instructive to consider several limiting scaling scenarios of Eq. 3.1. First, when $B \ll B_{\text{crit}}$, one obtains $\eta^*(T, B) \propto \eta_{\text{crit}}(T)/\sqrt{B_{\text{crit(T)}}}$, which we do not observe, since $\min(B_{\text{crit}}) = 2^{18}$ in our experiments. Second, when $B \gg B_{\text{crit}}$, one obtains $\eta^*(T, B) \propto \eta_{\text{crit}}(T) \cdot \sqrt{B_{\text{crit(T)}}}$. We observe this regime for high batch size values ($B = 2^{24}$ and $B = 2^{26}$) in the high token budget regime ($T > 2^{34}$ tokens). Since in this region, we see $\eta_{\text{crit}}(T) \sim 1$ (Fig. 2, right) and $B_{\text{crit}} \propto T^{\alpha_B} \approx T$, we obtain $\eta^* \propto \sqrt{T}$ (as can be seen in Fig. 1b). Lastly, a special case is when $B(T) = B_{\text{crit}}(T)$, i.e. batch size is tracking the critical one. We observe this regime for the smallest batch sizes ($B = 2^{16}$ and $B = 2^{18}$) in the low token budget regime ($T < 2^{33}$ tokens). In that case, the optimal learning rate $\eta^* \propto \eta_{\text{crit}} \propto T^{\alpha_\eta} = T^{-0.5}$ (see Fig. 2, where the power exponent $\alpha_\eta = -0.5$ describes data in the low token budget region better).

## 3.3 OPTIMALLY-TUNED BATCH SIZE INCREASES IN TIME

Second, we study how optimal hyperparameter values evolve in time to yield optimal loss values. For each batch size and horizon length, we select the best-performing run across the learning rate grid and plot model loss $\mathcal{L}_{\text{val}}$ against batch size across time horizons for the configuration with ($d_{\text{model}} = 1024$, $d_{\text{model}}^{\text{base}} = 1024$). Results are presented in Fig. 3, with a full set of plots across various combinations of ($d_{\text{model}}$, $d_{\text{model}}^{\text{base}}$) in Appendix A.12.

We observe an increase of the optimal batch size with increase of the pretraining token budget from $B^*|_{T=2^{30}} = 2^{18}$ to $B^*|_{T=2^{35}} = 2^{20}$ (Fig. 3a). Emergence of suboptimality is more pronounced when transposing the token budget and batch size axes (Fig. 3b), where the smallest $B = 2^{16}$ batch size curve, with each point having learning rate scaled approximately with the inverse scaling rule $\eta^* \propto 1/\sqrt{T}$, is being taken over in the $T \to \infty$ limit by the curves corresponding to larger batch sizes. Furthermore, comparing the optimal batch size values with the critical batch size evolution (Sec. 3.1), we obtain $B^*(T) < B_{\text{crit}}$ for the range of our measurements $T = [2^{30}, 2^{37}]$.

This result illustrates that, while naïve "pairwise" scaling rules for optimal learning rate, e.g. $\eta^* \propto 1/\sqrt{T}$, are convenient for predicting optimal values at scale, they do not necessarily result in the best model performance: taking batch size dynamics into account is required. In other words, the invariant induced solely by, for example, the $\eta^* \propto 1/\sqrt{T}$ scaling rule is not sufficient for the model performance to be optimal. We believe, similarly to Smith & Le (2018), that some broader notion of noise scale should serve as a more fundamental invariant to optimize for in the joint data and model size limit. We discuss this idea in more detail in Sec. 4.

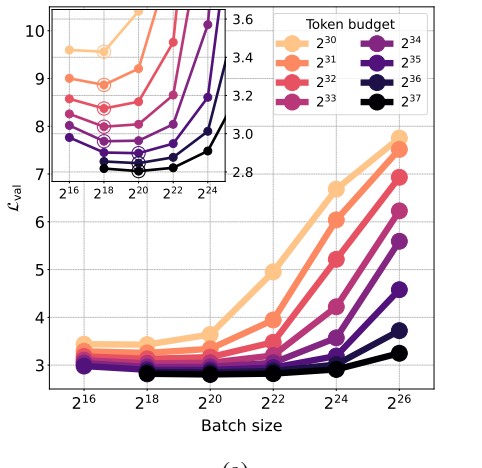 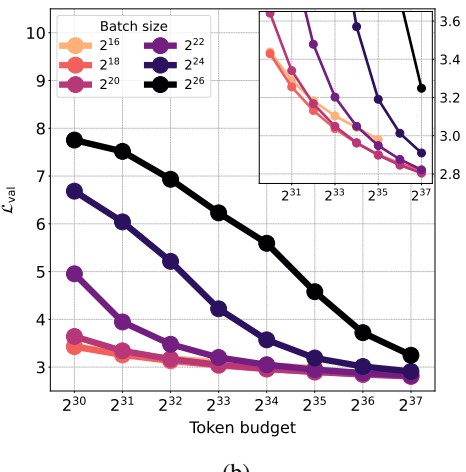

(a)                                          (b)

Figure 3: Validation loss $\mathcal{L}_{\text{val}}$ for a ($d_{\text{model}} = d_{\text{model}}^{\text{base}} = 1024$) model training (354M parameters) with an optimally-tuned learning rate as a function of **(a)** batch size split in pretraining token budgets **(b)** pretraining token budget split in batch size, both measured in tokens. Inset plots zoom into the optimum region. We observe that **(a)** optimal batch size (circled markers in the inset plot) evolves in time, by a $\times 2^2$ ($B = 2^{18} \to 2^{20}$ tokens) increase with an increase of the budget by $\times 2^5$ ($T = 2^{30} \to 2^{35}$ tokens) **(b)** smaller batch sizes are gradually phased out to become suboptimal as the token budget increases.

## 3.4 LEARNING RATE SENSITIVITY IS REDUCED IN TIME, AND IS UNCHANGED WITHIN $\mu$P

After having studied the learning rate optimum dynamics, we turn our attention to a broader structure around the optimum from the sensitivity perspective. Specifically, we are interested in how the *shape* of the $\mathcal{L}_{\text{val}}(\eta)$ curve changes in the time $T \to \infty$ and $\mu$P width limits. In Fig. 4, we present our observations for the two base models with $d_{\text{model}}^{\text{base}} = d_{\text{model}} \in \{256, 1024\}$, for token budgets $T \in \{2^{31}, 2^{33}, 2^{35}\}$. We note that since we implement $\mu$P in a way that the base model is also SP-parametrized, the results should be applicable to this parametrization as well.

We observe that there is a general decrease in the learning rate sensitivity by up to $2^1$ per each token budget increase by $2^2$ as measured by $\mathcal{L}_{\text{val}} - \mathcal{L}_{\text{val}}^{\min}$ value, where $\mathcal{L}_{\text{val}}^{\min} = \mathcal{L}_{\text{val}}(\eta^*)$ is the validation loss value in the learning rate optimum. This indicates that the model profits from longer training by having lower penalty for the misspecification of the optimal learning rate. Notably, the decrease is more pronounced for batch sizes in the critical region ($B = 2^{20}$ and $2^{22}$), while for the region with the $\eta^* \propto 1/\sqrt{T}$ scaling rule ($B = 2^{18}$), the effect is either reduced (base model $d_{\text{model}}^{\text{base}} = 1024$) or shows asymmetric trends w.r.t. the learning rate optimum (base model $d_{\text{model}}^{\text{base}} = 256$). However, within our measurement precision, the sensitivity evens out across batch sizes for the longest $2^{35}$ token horizon. Overall, our results motivate the choice of the training regime within the critical batch size region in order to minimize the risks of under- or overshooting the learning rate optimum. As we show in Appendix A.6, the learning rate optimum position can vary by a factor of two just depending on the random seed choice.

With respect to the $\mu$P width limit, we observe no significant deviation of the loss profile from the one of the base model, both for up- and down-scaled models within $\mu$P (Fig. 5 with and Fig. 21 without $\mathcal{L}_{\text{val}}$ normalization). Evaluated for the data horizon of $T = 2^{35} \approx 34$B tokens, this holds across the models with the number of trainable parameters ranging from 32M up to 5B. Likewise, changing the base model does not affect the profile shape, except for the optimum learning rate shift

---

[3]We note that the model is not fitted to the data points in the representation of Fig. 1b. Instead, as described in Sec. 3.1, the fit is performed in two consecutive steps: by first capturing the ($\eta^*$, $B$) behavior per token budget (illustrated in Fig. 1a), followed by fitting the time dynamics (illustrated in Fig. 2).

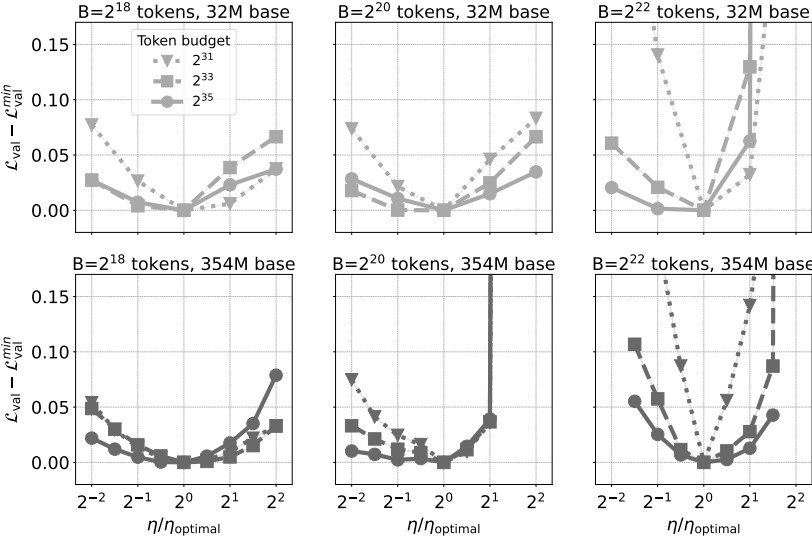

Figure 4: Learning rate sensitivity $\mathcal{L}_{\mathrm{val}} - \mathcal{L}_{\mathrm{val}}^{\min}$ as a function of the learning rate deviation from the optimal value $\eta/\eta_{\mathrm{optimal}}$, measured for batch sizes of $B = 2^{18}$ (left column), $2^{20}$ (middle column), and $2^{22}$ (right column) tokens, separately for the $\mu$P base models with width $d_{\mathrm{model}}^{\mathrm{base}} = 256$ (top row) and $1024$ (bottom row). The former model amounts to 32M and the latter to 354M trainable parameters. With an increase of the pretraining token budget (different marker styles) we observe a general decrease in the learning rate sensitivity, which is more pronounced for batch sizes $B \in \{2^{20}, 2^{22}\}$ in the critical region (Sec. 2.1) and for the 354M model. At the largest probed token budget $T = 2^{35}$ tokens, the sensitivity equalizes across the models and batch sizes.

by $\times 2$, which is expected for the base models compared here due to our $d_{\mathrm{model}}^{\mathrm{base}}$-dependent weight initialization scheme (Sec. 2.2).

# 4 DISCUSSION

While originally, we were aiming to find a golden recipe for hyperparameter transfer in the infinite data limit, we show that there is no simple and straight-forward answer. In Sec. 3.2, we show that the model fit based on Eq. 3.1 describes the observed data points well. However, as illustrated in Sec. 3.3, following the optimal learning rate trajectory in time is not sufficient to obtain optimal performance. That leads us to believe that there exists a deeper underlying perspective on the problem, as opposed to the one of simply tuning learning rate and batch size.

Fundamentally, the field of model parametrization research has originated from and is further converging towards preserving some notion of norm in some infinite (model width and/or depth) limit (Everett et al., 2024; Yang et al., 2024; Large et al., 2024). In fact, any parametrization itself is simply a set of scaling rules to be applied to hyperparameters in order to preserve these norms (e.g. of model weight matrices or weight updates). Expanding on this, one can argue that scaling rules follow from the requirement of keeping some underlying quantity invariant within the infinite limit. From this perspective, hyperparameter transfer is nothing but a consequence of such "conservation laws".

With this perspective in mind, we draw a parallel between infinite model and data limits, and speculate that a similar notion of "norm" should exist and should be aimed to be preserved in the infinite data limit. In fact, there is already a good candidate for this, namely the *noise scale* (Eq. 5 and 9), which intriguingly also induces scaling rules for hyperparameters (see Appendix A.2 for in-depth discussion). However, the existing definition neither takes into account the adaptive nature of the optimizer, nor the scenario of jointly following the infinite data and model limits.

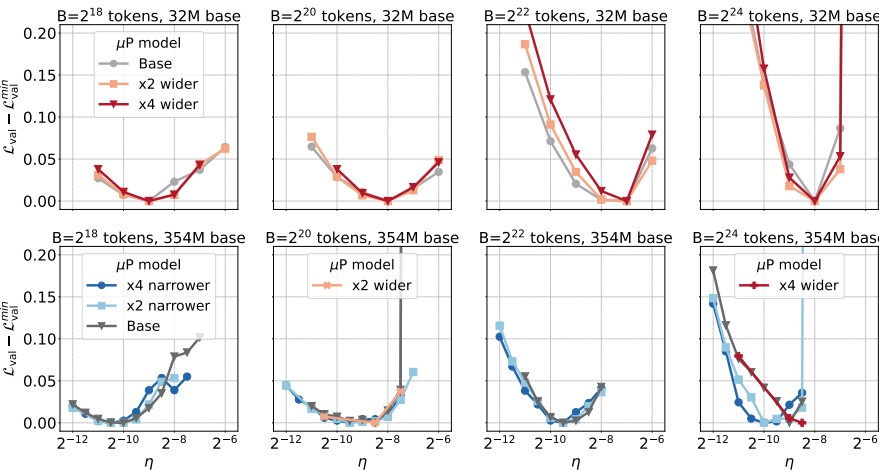

Figure 5: Learning rate sensitivity $\mathcal{L}_{\mathrm{val}} - \mathcal{L}_{\mathrm{val}}^{\min}$ as a function of learning rate $\eta$, measured for batch sizes of $B = 2^{18}$ (leftmost column), $2^{20}$ (middle left column), $2^{22}$ (middle right column) and $2^{24}$ (rightmost column) tokens, separately for the $\mu$P base models with the width $d_{\mathrm{model}}^{\mathrm{base}} = 256$ (top row) and 1024 (bottom row). Different marker styles correspond to different models within the $\mu$P family, with all the models being evaluated at the data horizon of $T = 2^{35}$ tokens. For the base model with $d_{\mathrm{model}}^{\mathrm{base}} = 256$, we scale the width only downwards, while for the base model with $d_{\mathrm{model}}^{\mathrm{base}} = 1024$, we scale it both upwards and downwards. We observe no significant difference in the sensitivity across all the $(d_{\mathrm{model}}^{\mathrm{base}}, d_{\mathrm{model}})$ configurations. Note that for the configuration ($B = 2^{24}$, $d_{\mathrm{model}}^{\mathrm{base}} = 1024$), the base and $d_{\mathrm{model}} = 4 \times d_{\mathrm{model}}^{\mathrm{base}}$ models share a different random seed compared to all the other models, to illustrate the loss penalty arising from the learning rate optimum variation.

We hope that our experimental observations, similarly to the discovery of the $\eta^* \propto T$ scaling rule for SGD, will make the first step towards the theoretical unification of infinite data and model size limits via deriving such a joint scaling invariant. Still, our insights into optimal scaling rules for learning rate and batch size might be valuable for practitioners who approach the problem of hyper-parameter optimization in the infinite data and model size limit. We provide our recommendations in Appendix A.1.

As future work, it would important to improve the resolution in data points with a finer grid of $(\eta, B)$ values. This is a necessary step to establish the generalization power of Eq. 3.1 and the power law fits of its critical parameters as a function of time (Fig. 2), also across various data sets, model architectures and modalities. Additionally, while we used $\mu P$ as the main way to incorporate model scaling due to its ability to transfer optimal learning rate across model sizes, recent work of Everett et al. (2024) suggests that this is not the only way to do so. A similar study to ours, but for other model parametrizations, is an exciting direction of future research.

## 5 RELATED WORK

$(\eta, B)$ **scaling rules** In efforts to accelerate model training, the $\eta \propto B$ rule for the SGD optimizer was found necessary to avoid performance loss due to increased batch size (Goyal et al., 2018), known as generalization gap (Keskar et al., 2017). Afterwards, additional usage of momentum (Smith et al., 2018) and model scaling (Park et al., 2019) was incorporated, and a $\eta \propto \sqrt{B}$ rule for Adam was observed (Hilton et al., 2022). From the theoretical side, experimentally observed rules were verified with the framework of stochastic differential equations (SDEs) (Smith & Le, 2018; Malladi et al., 2023), loss curvature analysis (Zhang et al., 2019; McCandlish et al., 2018; Li et al., 2024) and random matrix theory (Granziol et al., 2021). While most of the studies were performed in the fixed epoch budget, Shallue et al. (2019) broadened the perspective to other target budget

measures and studied the scope of the $\eta \propto B$ rule applicability across various datasets and model architectures. Looking beyond fixed budgets, Smith & Le (2018) showed a linear relation between the optimal batch size and the dataset size (for fixed $\eta$), and Smith et al. (2020) similarly presented hints for a linear relation between the optimal learning rate and the dataset size (for fixed $B$), with both works considering the SGD optimizer. In the modern LLM pretraining context, Hu et al. (2024); DeepSeek-AI et al. (2024) approached this problem by deriving the joint $(\eta, B)$ scaling laws.

**$\mu$P** Originally developed within the Tensor Program series studying feature learning in the infinite width limit (Yang & Hu, 2022; Yang et al., 2022), $\mu$P has been gaining traction recently within the LLM community. It has been extensively tested and applied experimentally (Lingle, 2024; Blake et al., 2024; Gunter et al., 2024; Dey et al., 2024), as well as theoretically, with Yang et al. (2023); Bordelon et al. (2024) extending it to the infinite depth limit, and Yang et al. (2024); Bernstein et al. (2023) revisiting it from the spectral normalization perspective. Recently, Everett et al. (2024) showed that other model parametrizations also induce hyperparameter transfer if taking weight alignment into account. Furthermore, they revealed that $\mu$Transfer does not work in the regime of Chinchilla-optimal scaling (Hoffmann et al., 2022). The most closely related work to ours, Shen et al. (2024) expanded on this observation and proposed a learning rate scheduler combining $\mu$P and experimentally measured $(\eta, B)$ scaling rules to allow for the hyperparameter transfer in the $T \to \infty$ limit, however only limited to the $\eta^* \propto 1/\sqrt{T}$ scaling regime.

**Sensitivity** The topic of loss sensitivity to suboptimal hyperparameter choice is less thoroughly studied, focusing exclusively on learning rate as the most affecting hyperparameter. Wortsman et al. (2023) studied how various optimizer and model interventions, such as weight decay or $\mu$P usage, influence the learning rate sensitivity with the model size scaling. Hägele et al. (2024) investigated the impact of various learning rate schedule choices, such as length and functional form of the decay phase.

## 6 CONCLUSION

In this work, we studied joint model and data scaling in the LLM context from the perspective of optimal learning rate $\eta$ and batch size $B$ dynamics. We observed an intricate dependence of optimal $\eta$ scaling on $B$ and its relation to the critical batch size $B_{\text{crit}}$, as a function of the pretraining token budget $T$. This dynamic is preserved during model scaling with $\mu$P, as well as the loss sensitivity to the learning rate variation, highlighting the intriguing difference in how $\mu$P infinite width and time limits evolve the critical batch size. Overall, we hope our observations pave the way towards deeper understanding of the optimal scaling in the unified infinite data and model size limit.

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

## A APPENDIX

### A.1 HYPERPARAMETER OPTIMIZATION IN THE INFINITE DATA AND MODEL SIZE LIMIT

We believe our observations provide useful hints on how to scale learning rate and batch size jointly in the infinite data and model size limits. We take the general $\mu$Transfer approach of tuning hyperparameters for a small proxy model and then transferring them either zero-shot or according to some scaling rules via extrapolation, across model sizes and data horizons.

1. If one can afford tuning a $\mu$P proxy model on the data horizon of the target model, then it is sufficient to simply perform a grid search over learning rate and batch size values to find the best combination, following $\mu$Transfer (Yang et al., 2022). As we describe in Sec. 5, $\mu$Transfer has been established to successfully transfer hyperparameters to O(10B) model sizes, albeit with potential limitations arising from very long range extrapolation in the infinite width limit (Blake et al., 2024; Gunter et al., 2024).

2. Otherwise, a proxy model has to be tuned on a shorter data horizon than the target one. In that case, we suggest running a 2D grid search across learning rate and batch size values roughly around the optimal ones, where each training follows a WSD schedule (Sec. 2.4), for as long as compute budget allows. We suggest both the warmup and decay of the schedule to be fixed to the one of the target model in *absolute number of tokens*, which in turn should be about 10–20% fraction of the target model horizon to be optimal (Kosson et al., 2024; Hägele et al., 2024). This is due to the observed drift of the learning rate optimum with the change of the number of steps (Appendix A.7). It is still not yet clear how scaling of warmup/decay length and Adam's $\beta_{1,2}$ parameters (which we keep constant in our experiments) can be incorporated into the total horizon scaling. We leave this as an interesting direction for future work.

3. After the grid search, one should be able to obtain a plot similar to Fig. 1a and Fig. 3a. Provided long enough WSD horizon, a drift in time of the critical batch size region, associated to the peak of the fixed token budget curve in Fig. 1a, should be visible. Likewise, there should be a drift of the optimally tuned (i.e. assuming optimal learning rate is used) batch size in time as in Fig. 3a. Since we observe a strong correlation but still a mismatch between the optimally-tuned batch size and the critical size, we suggest the following approach for selecting optimal hyperparameter values:

   (a) Derive scaling rule by extrapolating the batch size optimum drift in time $T$ based on Fig. 3a (in our case, approximately $B^* \propto \sqrt{T}$). Estimate the expected optimal batch size value $B^*_{\text{target}}$ for the target data horizon $T_{\text{target}}$ under assumption of the optimally tuned learning rate.

   (b) Perform a fit to fixed-budget curves per token budget step based on data similar to Fig. 1a with Eq. 3.1, following the procedure of Sec. 3.1. Fit a set of extracted $B_{\text{crit}}$ per

token budget with a power law function $B_{\text{crit}} = aT^{\alpha_B} + b$ to extract the corresponding exponent $\alpha_B$ (in our case, $\alpha_B \approx 1$) and derive the expected critical batch size for the target horizon $B_{\text{target}}^{\text{crit}}$.

(c) Perform the same power law fit to the $\eta^{\text{crit}}(T)$ data and extrapolate its value to the target horizon, obtaining $\eta_{\text{target}}^{\text{crit}}$. Set optimal learning rate for the target horizon as:

$$\eta_{\text{target}}^* = \begin{cases} \eta_{\text{target}}^{\text{crit}} \cdot \sqrt{B_{\text{target}}^*/B_{\text{target}}^{\text{crit}}} & \text{if } B_{\text{target}}^* \leq B_{\text{target}}^{\text{crit}} \\ \eta_{\text{target}}^{\text{crit}} \cdot \sqrt{B_{\text{target}}^{\text{crit}}/B_{\text{target}}^*} & \text{if } B_{\text{target}}^* > B_{\text{target}}^{\text{crit}} \end{cases}, \tag{3}$$

where we correct the learning rate value for the corresponding $\eta^*(B)$ scaling regime.

4. Apply optimal values of learning rate $\eta_{\text{target}}^*$ and batch size $B_{\text{target}}^*$ to the target model, scaled up with $\mu$P, and to the target training horizon. As we show in this work, $\mu$P does not impact the dynamics of the critical batch size evolution in the infinite data limit, therefore we expect no interference between the two limits.

We suppose it is also possible to adjust the recipe above to the continual learning setting (Çağatay Yıldız et al., 2024; Ibrahim et al., 2024): under assumption of $\eta^{\text{crit}}$ being constant in time and of the golden path hypothesis (Vyas et al., 2024), one could indefinitely run the model training with the same learning rate but dynamically adjust the batch size to follow the critical one (peak of the fixed budget curve in Fig. 1a), or, alternatively viewed, to remain on the pareto curve of Fig. 3b (inset plot).

## A.2 ON CRITICAL BATCH SIZE AND NOISE SCALE

There are two perspectives on the critical batch size $B_{\text{crit}}$. Firstly, McCandlish et al. (2018) define it as a batch size which results in an optimal trade-off between data sample efficiency and gradient update step efficiency:

$$B_{\text{crit}} := \frac{E_{\min}}{S_{\min}}, \tag{4}$$

where $E_{\min}$ ($S_{\min}$) are the minimum possible number of training examples (steps) to reach a specified level of performance. Additionally, they introduce a notion of a *noise scale* (for SGD-like optimizers):

$$B_{\text{noise}}^{\text{curv}} := \frac{tr(H\Sigma)}{G^T H G}, \tag{5}$$

where $G$ is the noiseless true gradient, $H$ is the true hessian of the loss function and $\Sigma$ is the minibatch covariance. For $B \ll B_{\text{noise}}^{\text{curv}}$ one obtains the linear learning rate scaling rule, while for $B \gg B_{\text{noise}}^{\text{curv}}$ increasing $B$ does not yield any loss improvement.

Under assumption of the Hessian being a multiple of the identity matrix, one obtains a simplified version:

$$B_{\text{simple}}^{\text{curv}} := \frac{tr(\Sigma)}{|G^2|}, \tag{6}$$

and McCandlish et al. (2018) argue that

$$B_{\text{crit}} \approx B_{\text{noise}}^{\text{curv}} \propto B_{\text{simple}}^{\text{curv}}, \tag{7}$$

thus bridging together mathematical loss curvature and pragmatical compute resource utilization views. Approximation with $B_{\text{simple}}^{\text{curv}}$, being computationally less expensive to estimate, is shown to be to a good degree applicable across multiple tasks, datasets and model architectures. Both the

critical batch size and the noise scale are shown to grow in time as one progresses in the training, with the only dependence on the loss value via a power law, with parameters $B_0$ and $\alpha_B$ to be determined empirically (Kaplan et al., 2020):

$$B_{\text{crit}} = \frac{B_0}{L^{1/\alpha_B}}. \tag{8}$$

Notably, Smith & Le (2018) introduce from a different SDE perspective another definition of the noise scale:

$$B_{\text{noise}}^{\text{SDE}} := \eta\left(\frac{T}{B} - 1\right) \approx \eta\frac{T}{B}, \tag{9}$$

where $T$ is the training set size. It is suggested that one should aim at finding the optimal noise scale in the first place, rather than optimal batch size and learning rate. Within the suggested Bayesian framework, Smith & Le (2018) argue that the optimality arises from the trade-off between depth and breadth in the Bayesian evidence. In a follow-up work, Park et al. (2019) take one step further and extend the noise scale to a model width limit and introduce a modified noise scale accounting for the change of the model width in the standard (SP) and Neural Tangent Kernel (NTK) parametrizations (Jacot et al., 2020):

$$B_{\text{noise}}^{\text{norm}} := \frac{B_{\text{noise}}^{\text{SDE}}}{|w|^2}, \tag{10}$$

where $|w|^2$ is model weight norm, normalizing $B_{\text{noise}}^{\text{SDE}}$ to have the unit $1/\text{loss}$.

The second perspective on $B_{\text{crit}}$ is as a region where *batch invariance* breaks. Introduced by Hilton et al. (2022), batch invariance refers to a regime where the model performance remains invariant with the change of either learning rate or batch size within the corresponding scaling rule. As shown by Shallue et al. (2019), the breaking of batch invariance appears with an increase of batch size to sufficiently large values and looks like plateauing of the optimal learning rate. Zhang et al. (2019) further investigated how the critical batch size is affected by using momentum, optimizer pre-conditioning and exponential moving average (EMA).

Intriguingly, Li et al. (2024) expanded the approach of McCandlish et al. (2018) and showed that in the case of Adam, the batch invariance does not break conventionally as in the SGD case. In fact, it is always preserved, with the only difference that the $\eta \propto \sqrt{B}$ scaling rule breaks at the peak value $B_{\text{peak}}$ and transforms into a $\eta \propto 1/\sqrt{B}$ rule via:

$$\eta^* = \frac{\eta^{\text{crit}}}{\frac{1}{2}\left(\sqrt{\frac{B_{\text{peak}}}{B}} + \sqrt{\frac{B}{B_{\text{peak}}}}\right)}. \tag{11}$$

They also show that $B_{\text{peak}} \approx B_{\text{crit}}$ in the definition of McCandlish et al. (2018), therefore bridging together the two $B_{\text{crit}}$ perspectives outlined above.

### A.3 MODEL TRAINING CONFIGURATION (CONT.)

- 24 layers, FFN expansion factor $f_{\text{ffn}} = d_{\text{ffn}}/d_{\text{model}} = 4$, multihead attention with the head dimension $d_{\text{head}} = 128$.
- GeLU activation function, Layer Normalization initialized with 1 (Ba et al., 2016), RoPE with $\theta = 10000$ (Su et al., 2023).
- Dropout is disabled and biases are included in all layers (initialized with 0), weights are shared between the input and output embedding layers.
- FSDP parallelization scheme (Zhao et al., 2023), bfloat16 precision, FlashAttention-2 (Dao, 2023).

## A.4 HYPERPARAMETER GRID (CONT.)

The $(\eta, B, T, d_{\text{model}}^{\text{base}}, d_{\text{model}})$ grid is defined with the following values:

- Learning rate $\eta$:
    - $\{2^{-12}, 2^{-11.5}, \ldots, 2^{-7}\}$ for $d_{\text{model}}^{\text{base}} = 1024$
    - $\{2^{-11}, 2^{-10}, \ldots, 2^{-6}\}$ for $d_{\text{model}}^{\text{base}} = 256$
- Batch size $B = \{2^{16}, 2^{18}, \ldots, 2^{26}\}$ tokens
- Data horizon $T = \{2^{30}, 2^{31} \ldots, 2^{35}\}$ tokens
- Base model width $d_{\text{model}}^{\text{base}} = \{256, 1024\}$
- Model width $d_{\text{model}} = \{256, 512, 1024\}$

For a configuration with $(B = 2^{20}, d_{\text{model}}^{\text{base}} = 1024)$, we perform longer runs with an extended set of horizons with $\{2^{36}, 2^{37}\}$ token budgets, except for the smallest $B = 2^{16}$ due to limited computational resources and low GPU utilization of this batch size on our hardware. A configuration for the largest batch size $(B = 2^{26}, d_{\text{model}}^{\text{base}} = 1024)$, we train until $T = 2^{38}$ tokens to further establish the learning rate optimum drift (Sec. 3.2).

The total number of trainable parameters is 32M, 101M, 354M for the models with widths $d_{\text{model}} = \{256, 512, 1024\}$, respectively. We also train 1.3B and 5B models up until $2^{35} \approx 34$B tokens with three selected learning rate values for a fixed batch size of $2^{20}$ and $2^{24}$ tokens, respectively, in order to study learning rate sensitivity change within $\mu$P (Sec. 3.4). The models share the same $\mu$P base model with $d_{\text{model}}^{\text{base}} = 1024$ and have the corresponding width $d_{\text{model}} = 2048$ (1.3B) and $d_{\text{model}} = 4096$ (5B).

## A.5 FITTING PROCEDURE

In Sec. 3.1, we introduce the fitting procedure to the data points of Fig. 1a with a functional form of Eq. 3.1 with two parameters $\eta_{\text{crit}}$ and $B_{\text{crit}}$. We perform the fit separately per token budget, as illustrated in Fig. 6, using `scipy.optimize.curve_fit()` and firstly without including error bars (which stem from the variation of the model with $\mu P$ and random seeds). Since we observe some data points as having no uncertainties, which makes the fit computationally unstable, we repeat the fit two more times: one with adding small $\epsilon = 10^{-15}$ as uncertainty for such data points. Then, we attribute the mean uncertainty across the other points to the points without uncertainties and perform the same fit. This procedure results in three data sets for each $\eta_{\text{crit}}(T)$ and $B_{\text{crit}}(T)$, corresponding to three variations of the fitting procedure, which we treat as "systematic" uncertainty. For each data point, we assign an uncertainty as a square root of the corresponding covariance matrix element, as obtained from the fit.

We then perform a full power law $p_{\text{crit}} = a_p T^{\alpha_p} + b_p, \ p \in \{\eta, B\}$ fit with uncertainties to each of the three data sets, for each of the critical parameters $p$. We obtain the following values with corresponding fit uncertainties:

- No error:
    - $(a_\eta, \alpha_\eta, b_\eta) = (1.9 \cdot 10^5, -0.85, 2.9 \cdot 10^{-3}]$
    - $(a_B, \alpha_B, b_B) = [8.2 \cdot 10^{-5}, 1.00, 3.0 \cdot 10^5)$
    - $(\sigma_{a_\eta}, \sigma_{\alpha_\eta}, \sigma_{b_\eta}) = (1.2 \cdot 10^6, 0.32, 2.8 \cdot 10^{-4})$
    - $(\sigma_{a_B}, \sigma_{\alpha_B}, \sigma_{b_B}) = (4.5 \cdot 10^{-4}, 0.23, 1.5 \cdot 10^5)$
- With errors + $\epsilon$:
    - $(a_\eta, \alpha_\eta, b_\eta) = (2.3 \cdot 10^9, -1.31, 2.7 \cdot 10^{-3})$
    - $(a_B, \alpha_B, b_B) = (2.3 \cdot 10^{-2}, 0.75, 1.9 \cdot 10^5)$
    - $(\sigma_{a_\eta}, \sigma_{\alpha_\eta}, \sigma_{b_\eta}) = (1.0 \cdot 10^{10}, 0.21, 1.1 \cdot 10^{-4})$
    - $(\sigma_{a_B}, \sigma_{\alpha_B}, \sigma_{b_B}) = (5.1 \cdot 10^{-2}, 0.09, 8.8 \cdot 10^4)$
- With errors + mean uncertainty attribution:
    - $(a_\eta, \alpha_\eta, b_\eta) = (4.5 \cdot 10^{12}, -1.68, 2.9 \cdot 10^{-3})$

- $(a_B, \alpha_B, b_B) = (4.9 \cdot 10^{-7}, 1.20, 2.8 \cdot 10^5)$
- $(\sigma_{a_\eta}, \sigma_{\alpha_\eta}, \sigma_{b_\eta}) = (4.5 \cdot 10^{13}, 0.47, 1.4 \cdot 10^{-4})$
- $(\sigma_{a_B}, \sigma_{\alpha_B}, \sigma_{b_B}) = (2.1 \cdot 10^{-6}, 0.17, 8.0 \cdot 10^4)$

Since we are primarily interested in the power exponents $\alpha_p$, we average their values across the three variations to produce a central value for each $p$. For the uncertainties, we add in quadratures the variance across the three fit variations and the mean uncertainty obtained from each of the individual fits. This produces the results we supply in the main text (Eq. 3.1). To further constrain large uncertainties on $a_p$, we refit $\eta_{\mathrm{crit}}(T)$ and $B_{\mathrm{crit}}(T)$ time dependence data with the power exponents $\alpha_p$ fixed to the ones obtained above. This gives us the final model parameters, which we visualize oi Fig. 2 and 1b and discuss throughout the main text:

- $a_\eta = (2.0 \pm 0.3) \cdot 10^9$
- $\alpha_\eta = -1.3 \pm 0.4$
- $b_\eta = (3.1 \pm 0.1) \cdot 10^{-3}$
- $a_B = (8.0 \pm 1.3) \cdot 10^{-5}$
- $\alpha_B = 1.0 \pm 0.2$
- $b_B = (3.0 \pm 1.1) \cdot 10^5$

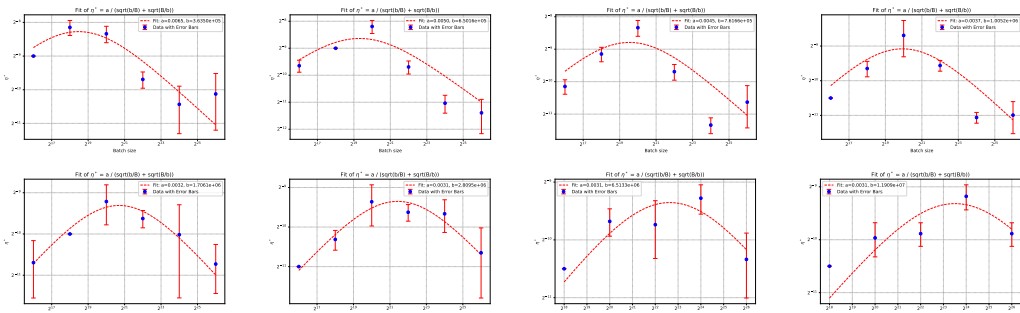

Figure 6: Fits to token budgets $T = 2^{30}, 2^{31}, \ldots, 2^{37}$ (from upper left to bottom right) with Eq. 3.1 to the data points in Fig. A.9.

## A.6 RANDOM SEED VARIATION

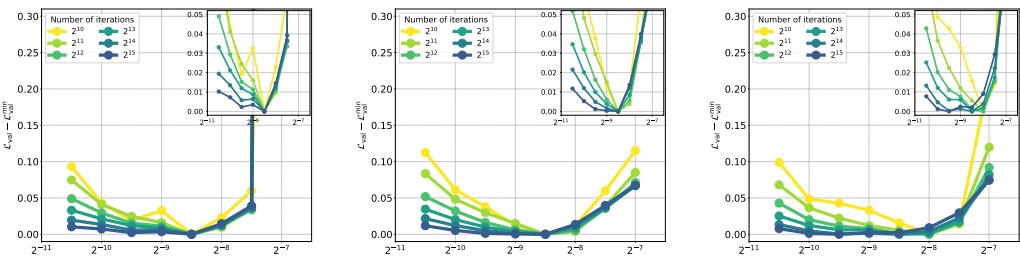

Figure 7: Loss profile $\mathcal{L}_{\mathrm{val}} - \mathcal{L}_{\mathrm{val}}^{\min}$ as a function of maximum learning rate $\eta$ for three different random seeds for the model configuration ($d_{\mathrm{model}} = d_{\mathrm{model}}^{\mathrm{base}} = 1024$).

## A.7 LEARNING RATE SCHEDULE SCALING (CONT.)

Conventionally, the learning rate schedule consists of a warmup phase, followed by either a constant phase or a decay phase. When all of the three phases are enabled, one obtains a warmup-stable-

decay (WSD) schedule (Hu et al., 2024):

$$\eta(t) = \begin{cases} \dfrac{t}{T_{\text{warmup}}} \cdot \eta_{\max} & \text{if } t < T_{\text{warmup}} \\ \eta_{\max} & \text{if } T_{\text{warmup}} \leq t < T - T_{\text{decay}} , \\ \left(1 - \dfrac{t - (T - T_{\text{decay}})}{T_{\text{decay}}}\right) \cdot \eta_{\max} & \text{if } T - T_{\text{decay}} \leq t < T \end{cases} \tag{12}$$

where $T$ is the total length of the training horizon, $T_{\text{warmup}}$ ($T_{\text{decay}}$) is the length of the warmup (decay) phases, all measured in tokens.

As Hägele et al. (2024) showed, there is no significant difference in terms of the final loss value and learning rate sensitivity between using cosine decay and WSD schedules. We run additional ablations in our setup and also arrive at the same conclusions: the structure of the learning rate optimum is marginally affected by the decay phase of the schedule and its type. Even though there appears to be a small increase in learning rate sensitivity if learning rate is decayed comparing to the schedule without decay, it does not affect the optimal $\eta^*$ location (Fig. 8).

Furthermore, we vary the warmup scaling strategy with an increase of the data horizon, specifically where all the horizons either share the same warmup length, or warmup is scaled together with the horizon length (with the fixed $f = T_{\text{warmup}}/T = 1/64$ fraction of the total horizon), or warmup is disabled. We observe that the addition of warmup decreases learning rate sensitivity and, interestingly, that scaling of the warmup proportionally with the horizon length leads to a drift of the learning rate optimum, as also indirectly observed earlier by Kosson et al. (2024).

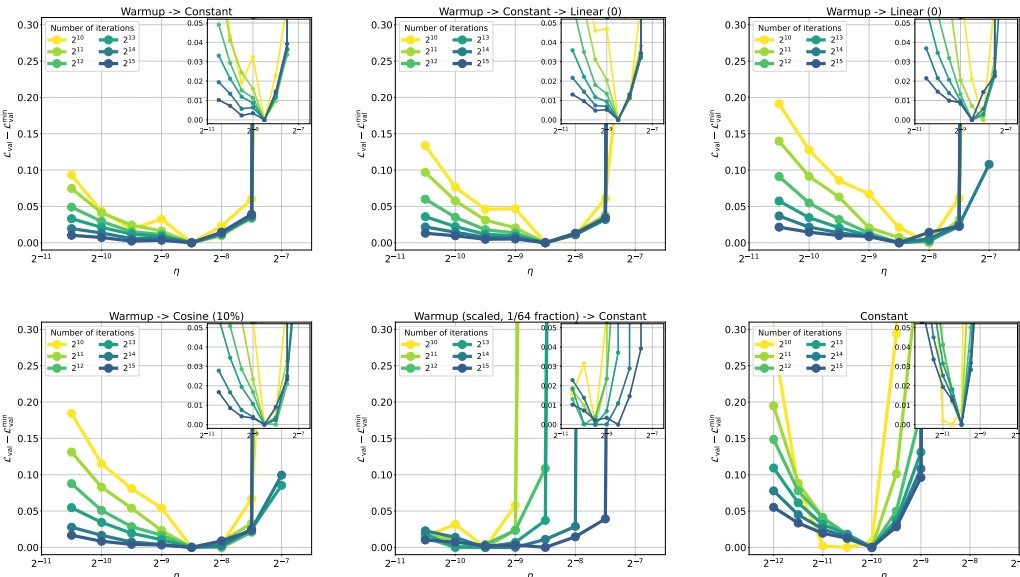

Figure 8: Loss profile $\mathcal{L}_{\text{val}} - \mathcal{L}_{\text{val}}^{\min}$ as a function of maximum learning rate $\eta$ for schedules with the following phases: warmup and constant (top left); warmup, constant and linear decay to 0 (top middle); warmup and linear decay to 0 (top right); warmup and cosine decay to 10% of the maximum $\eta$ (bottom left); warmup scaled as $1/64$ fraction of the total horizon and constant (bottom middle); constant (bottom right). Warmup duration is always set to $T_{\text{warmup}} = 2^{19} = 524288$ tokens, except for the case with warmup phase scaling. The model configuration is ($d_{\text{model}}^{\text{base}} = 1024$, $d_{\text{model}} = 1024$, $B = 2^{20}$).

## A.8 Loss profiles per $(d_{\text{model}}^{\text{base}}, d_{\text{model}})$ configuration

### A.8.1 $d_{\text{model}}^{\text{base}} = 1024$, $d_{\text{model}} = 1024$

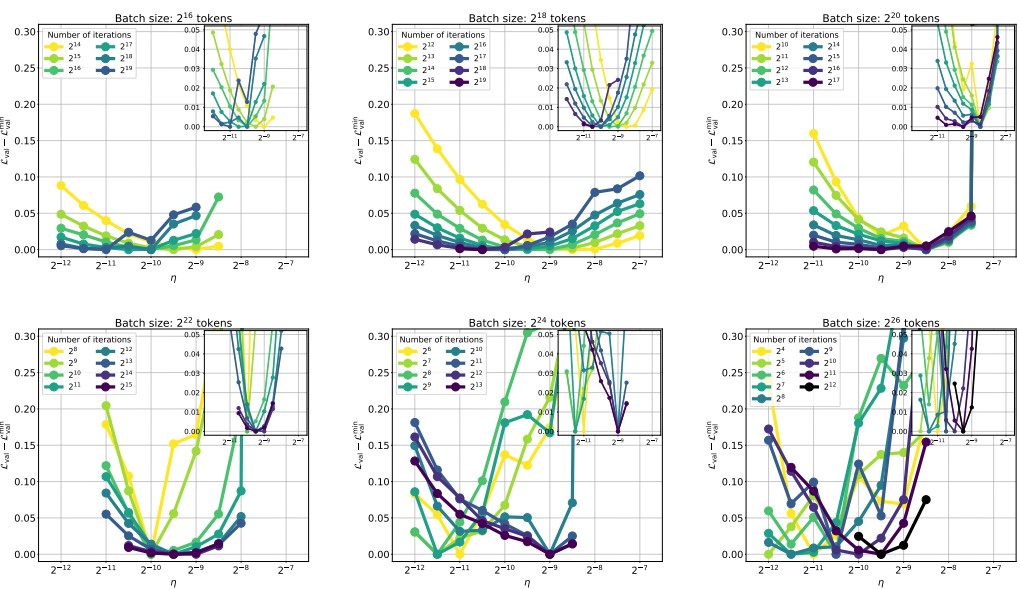

Figure 9: Loss profile $\mathcal{L}_{\text{val}} - \mathcal{L}_{\text{val}}^{\min}$ as a function of maximum learning rate $\eta$ for $(d_{\text{model}}^{\text{base}} = 1024$, $d_{\text{model}} = 1024)$ for batch size $B = 2^{16}$ (top left), $B = 2^{18}$ (top middle), $B = 2^{20}$ (top right), $B = 2^{22}$ (bottom left), $B = 2^{24}$ (bottom middle), $B = 2^{26}$ (bottom right) across various token budgets.

## A.8.2 $d_{\mathrm{model}}^{\mathrm{base}} = 1024$, $d_{\mathrm{model}} = 512$

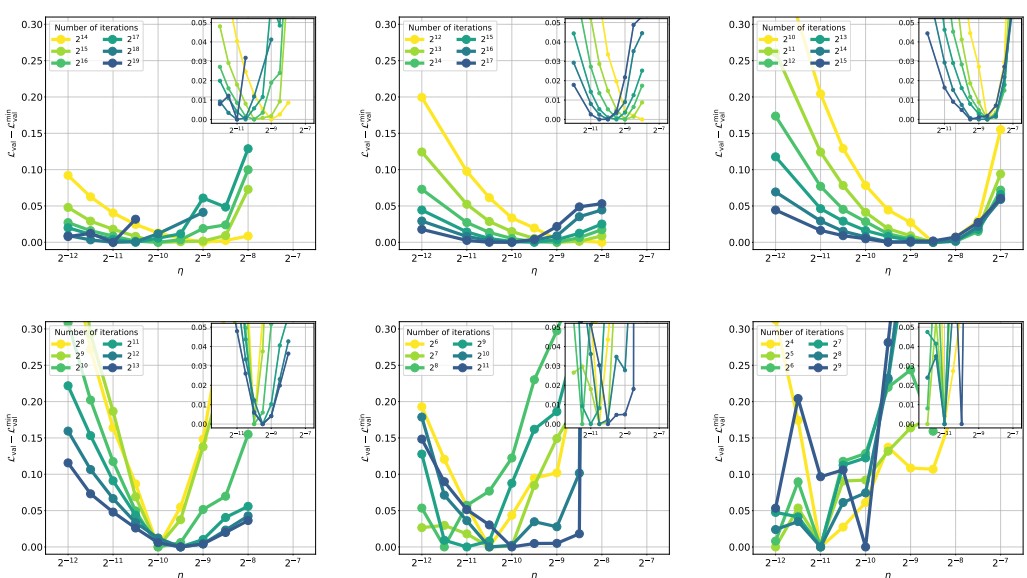

Figure 10: Loss profile $\mathcal{L}_{\mathrm{val}} - \mathcal{L}_{\mathrm{val}}^{\mathrm{min}}$ as a function of maximum learning rate $\eta$ for ($d_{\mathrm{model}}^{\mathrm{base}} = 1024$, $d_{\mathrm{model}} = 512$) for batch size $B = 2^{16}$ (top left), $B = 2^{18}$ (top middle), $B = 2^{20}$ (top right), $B = 2^{22}$ (bottom left), $B = 2^{24}$ (bottom middle), $B = 2^{26}$ (bottom right) across various token budgets.

## A.8.3 $d_{\mathrm{model}}^{\mathrm{base}} = 1024$, $d_{\mathrm{model}} = 256$

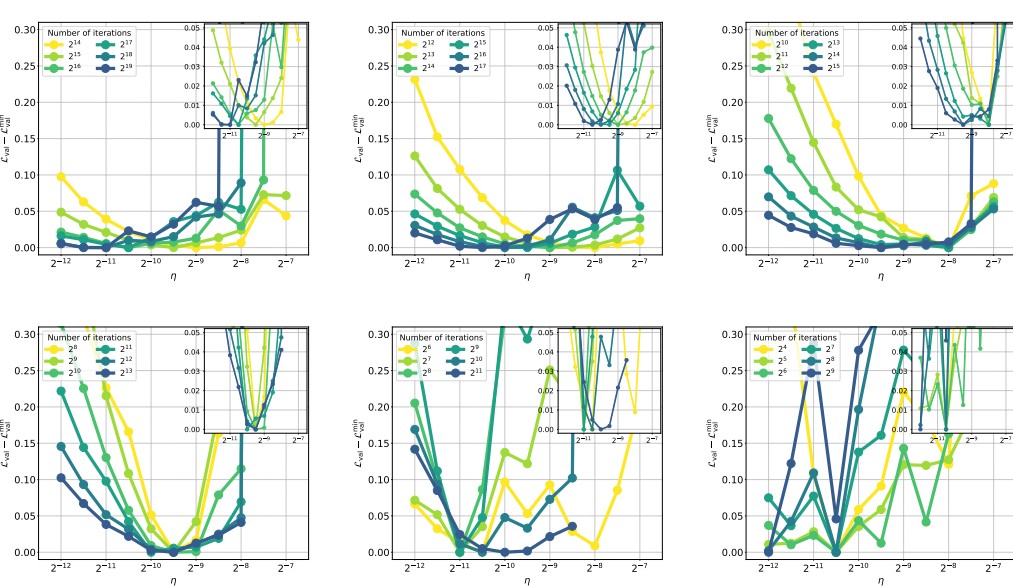

Figure 11: Loss profile $\mathcal{L}_{\mathrm{val}} - \mathcal{L}_{\mathrm{val}}^{\mathrm{min}}$ as a function of maximum learning rate $\eta$ for ($d_{\mathrm{model}}^{\mathrm{base}} = 1024$, $d_{\mathrm{model}} = 256$) for batch size $B = 2^{16}$ (top left), $B = 2^{18}$ (top middle), $B = 2^{20}$ (top right), $B = 2^{22}$ (bottom left), $B = 2^{24}$ (bottom middle), $B = 2^{26}$ (bottom right) across various token budgets.

### A.8.4 $d_{\text{model}}^{\text{base}} = 256$, $d_{\text{model}} = 256$

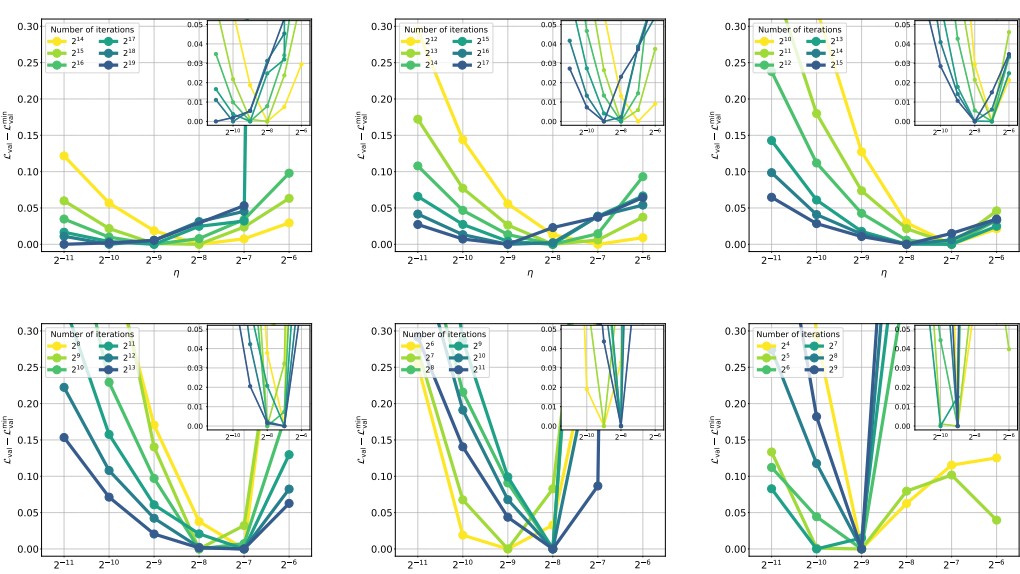

Figure 12: Loss profile $\mathcal{L}_{\text{val}} - \mathcal{L}_{\text{val}}^{\text{min}}$ as a function of maximum learning rate $\eta$ for ($d_{\text{model}}^{\text{base}} = 256$, $d_{\text{model}} = 256$) for batch size $B = 2^{16}$ (top left), $B = 2^{18}$ (top middle), $B = 2^{20}$ (top right), $B = 2^{22}$ (bottom left), $B = 2^{24}$ (bottom middle), $B = 2^{26}$ (bottom right) across various token budgets.

### A.8.5 $d_{\text{model}}^{\text{base}} = 256$, $d_{\text{model}} = 512$

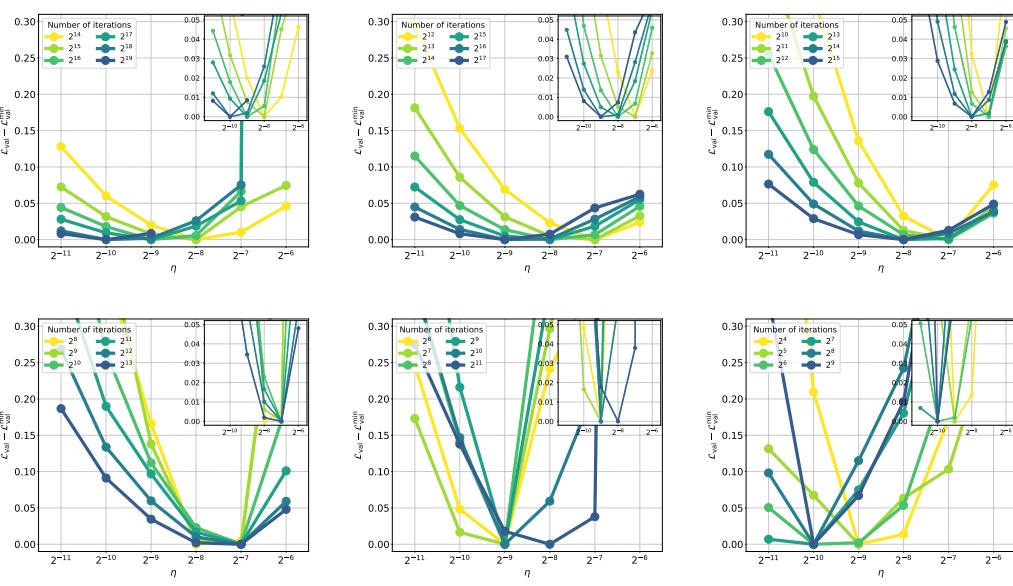

Figure 13: Loss profile $\mathcal{L}_{\text{val}} - \mathcal{L}_{\text{val}}^{\text{min}}$ as a function of maximum learning rate $\eta$ for ($d_{\text{model}}^{\text{base}} = 256$, $d_{\text{model}} = 512$) for batch size $B = 2^{16}$ (top left), $B = 2^{18}$ (top middle), $B = 2^{20}$ (top right), $B = 2^{22}$ (bottom left), $B = 2^{24}$ (bottom middle), $B = 2^{26}$ (bottom right) across various token budgets.

A.8.6 $d_{\text{model}}^{\text{base}} = 256,\ d_{\text{model}} = 1024$

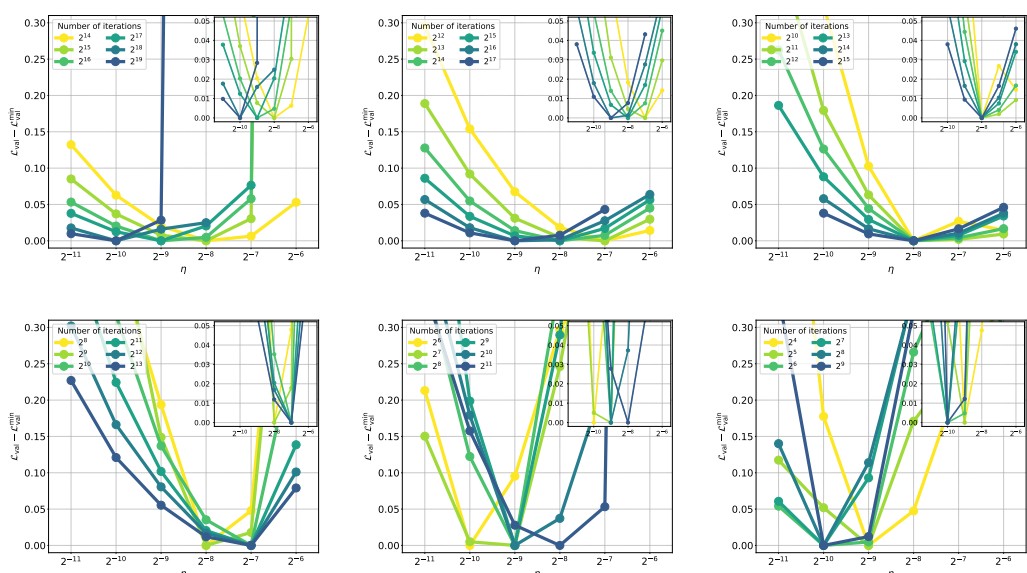

Figure 14: Loss profile $\mathcal{L}_{\text{val}} - \mathcal{L}_{\text{val}}^{\text{min}}$ as a function of maximum learning rate $\eta$ for $(d_{\text{model}}^{\text{base}} = 256,\ d_{\text{model}} = 1024)$ for batch size $B = 2^{16}$ (top left), $B = 2^{18}$ (top middle), $B = 2^{20}$ (top right), $B = 2^{22}$ (bottom left), $B = 2^{24}$ (bottom middle), $B = 2^{26}$ (bottom right) across various token budgets.

## A.9 Fig. 1 with the full set of batch size and token budget values

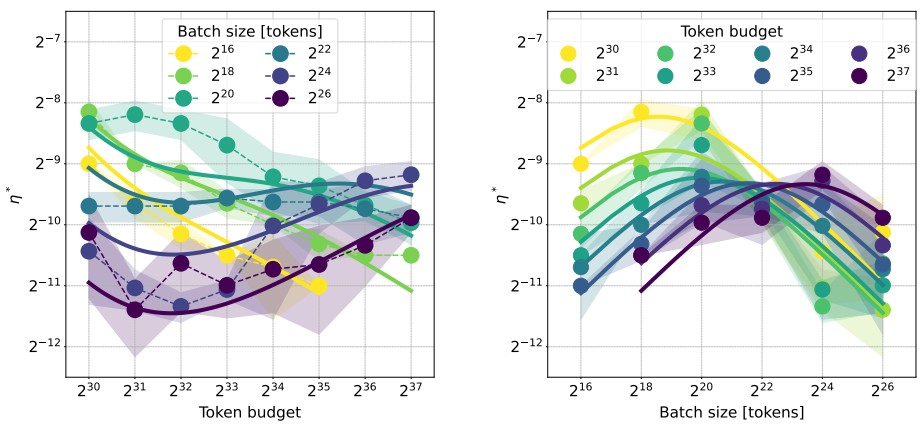

Figure 15: Same as Fig. 1 with the full set of batch size (left) and token budget (right) values.

## A.10 $\mu$P-AVERAGED OPTIMAL LEARNING RATE AND BATCH SIZE JOINT SCALING FOR $d_{\text{model}}^{\text{base}} = 256$

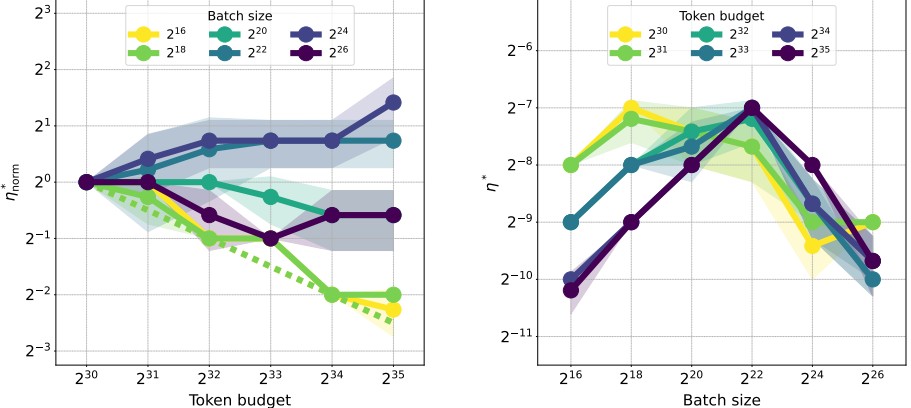

Figure 16: (left) Evolution of the optimal learning rate with an increase of the pretraining token budget $\eta_{\text{norm}}^*(T)$, normalized to $\eta^*|_{T=2^{30}}$, for a set of batch sizes (in tokens). Each point is obtained by averaging optimal learning rate values across $\mu$P model family, as described in Sec. 3.2. Dashed lines correspond to a square-root $\eta^* \propto \sqrt{T^{-1}}$ scaling rule. (right) Transposition of (left): optimal learning rate $\eta^*$ per batch size, against a range of pretraining token budgets. Each point is $\mu$P-averaged as in (left), with color bands visualizing the corresponding standard deviation. We note that experiments were performed with a coarser learning rate resolution of $2^1$ compared to a $2^{0.5}$ step in experiments with $d_{\text{model}}^{\text{base}} = 1024$.

## A.11 PER-MODEL OPTIMAL LEARNING RATE AND BATCH SIZE JOINT SCALING

### A.11.1 $d_{\mathrm{model}}^{\mathrm{base}} = 256$

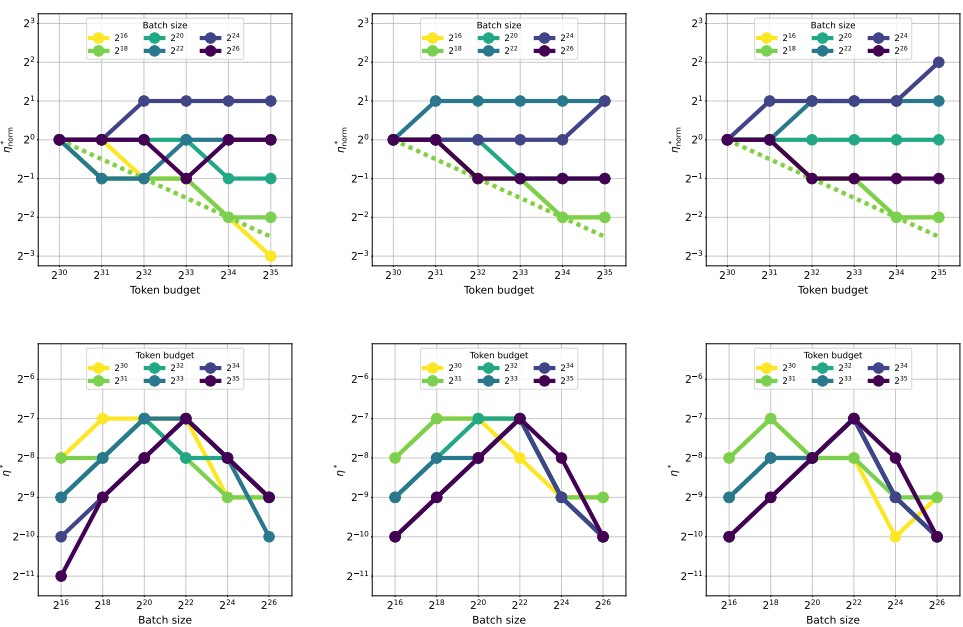

Figure 17: Individual curves contributing to Fig. A.10 for models with $d_{\mathrm{model}} = 256$ (left column), 512 (middle column), 1024 (right column) showing evolution of the normalized to $T = 2^{30}$ tokens optimal learning rate $\eta_{\mathrm{norm}}^{*}$ in time per batch size (top row), and joint optimal $(\eta, B)$ curves per token budget (bottom row), for $d_{\mathrm{model}}^{\mathrm{base}} = 256$.

A.11.2   $d_{\mathrm{model}}^{\mathrm{base}} = 1024$

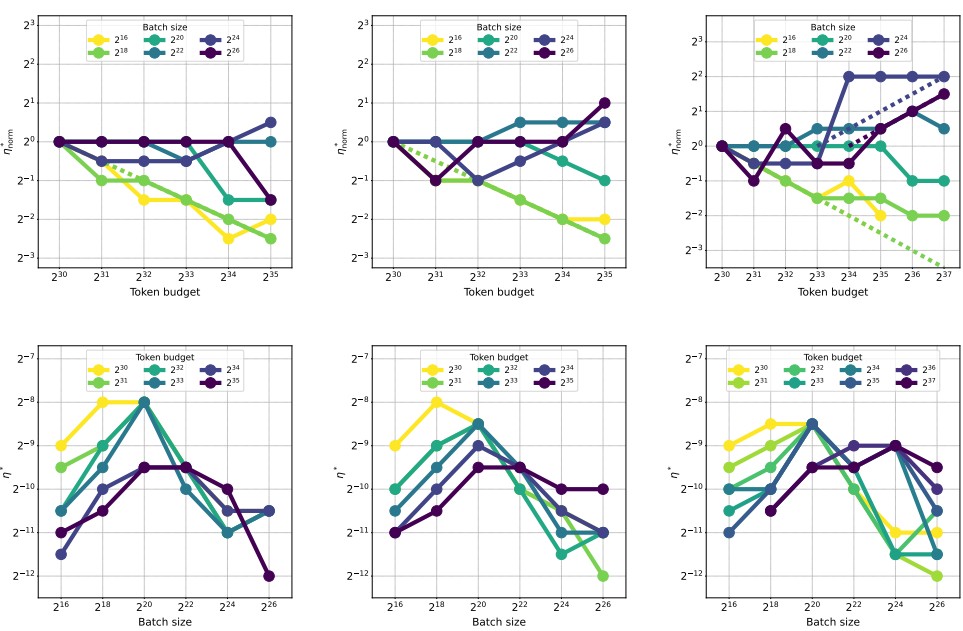

Figure 18: Individual curves contributing to Fig. 1 for models with $d_{\mathrm{model}} = 256$ (left column), 512 (middle column), 1024 (right column) showing evolution of the normalized to $T = 2^{30}$ tokens optimal learning rate $\eta_{\mathrm{norm}}^*$ in time per batch size (top row), and joint optimal $(\eta, B)$ curves per token budget (bottom row), for $d_{\mathrm{model}}^{\mathrm{base}} = 1024$.

## A.12 PER-MODEL VALIDATION LOSS EVOLUTION IN TIME DEPENDING ON BATCH SIZE WITH OPTIMALLY-TUNED LEARNING RATE

### A.12.1 $d_{\mathrm{model}}^{\mathrm{base}} = 256$

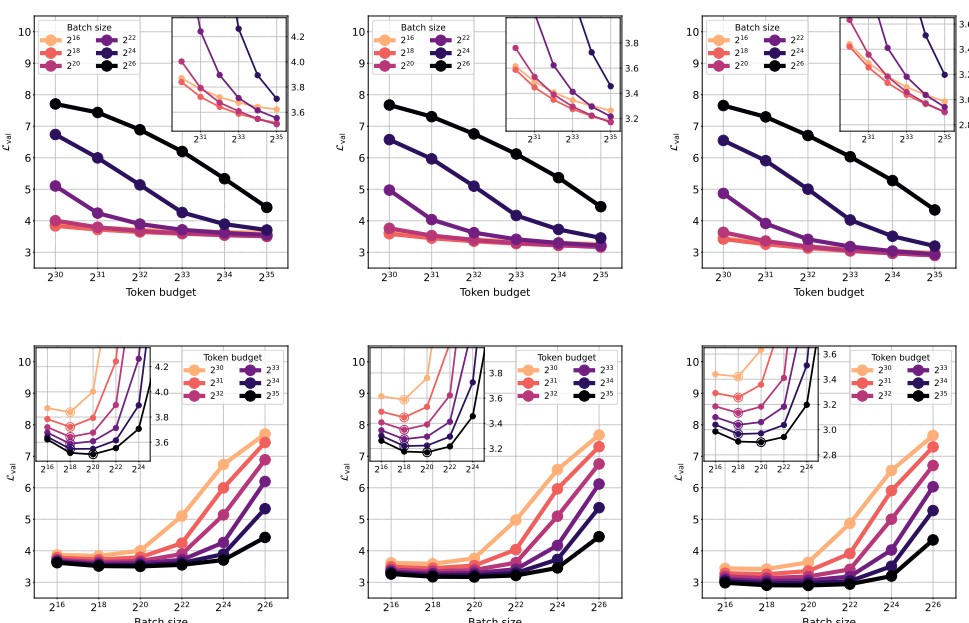

Figure 19: Analogue of Fig. 3b (top row) and Fig. 3a (bottom row) for models with widths $d_{\mathrm{model}} = 256$ (left column), $512$ (middle column), $1024$ (right column) and the base model width $d_{\mathrm{model}}^{\mathrm{base}} = 256$.

## A.12.2 $d_{\text{model}}^{\text{base}} = 1024$

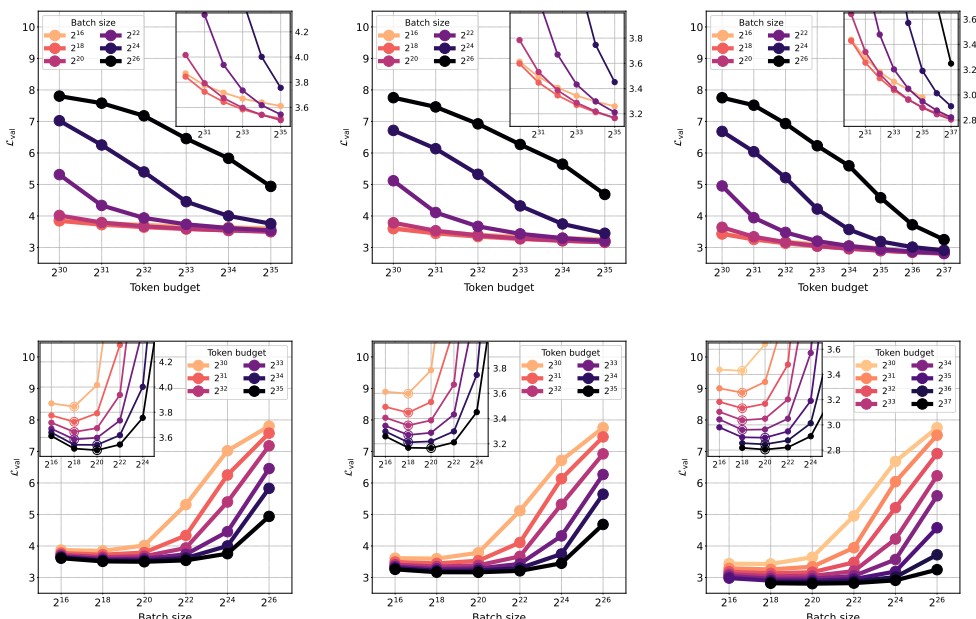

Figure 20: Analogue of Fig. 3b (top row) and Fig. 3a (bottom row) for models with widths $d_{\text{model}} = 256$ (left column), $512$ (middle column), $1024$ (right column) and the base model width $d_{\text{model}}^{\text{base}} = 1024$.

## A.13 LEARNING RATE SENSITIVITY IN THE $\mu$P WIDTH LIMIT

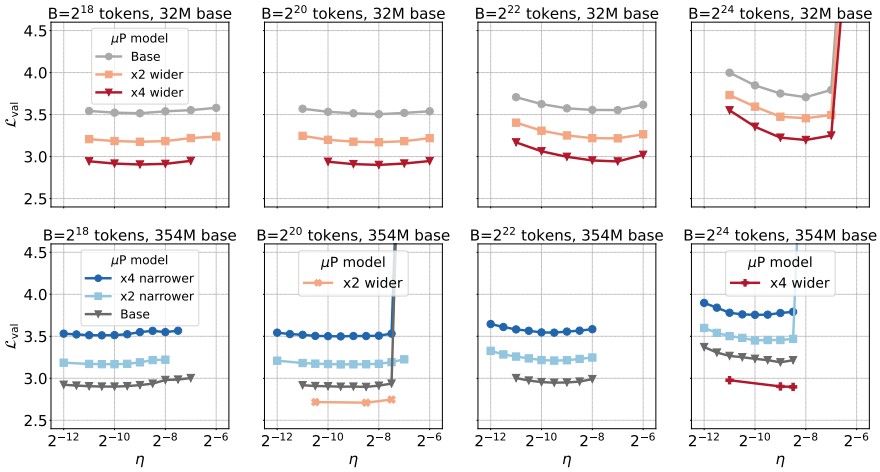

Figure 21: Same as Fig. 5 but without y-axis normalization with $\mathcal{L}_{\text{val}}^{\min}$.

