# OpenReview forum: "Time Transfer: On Optimal Learning Rate and Batch Size In The Infinite Data Limit"
_ICLR.cc/2025/Conference — Submitted to ICLR 2025_

### Official Review · Reviewer_C7LK · 2024-11-04

**Soundness:** 2
**Presentation:** 2
**Contribution:** 2
**Rating:** 5
**Confidence:** 3

**Summary:**

This paper investigates the optimal learning rate $\eta$ and batch size $B$ as a function of the data size $T$ in language model (LM) training, using the maximal update parametrization (muP). While muP focuses on hyperparameter transfer as model width increases (with other parameters fixed), this work centers on how hyperparameters evolve as $T$ grows, assuming a large but fixed model size (with results averaged across three model sizes). The authors show that the optimal $(\eta, B)$ can vary in complex ways with $T$; for example, there are regions where $\eta \propto 1/\sqrt{T}$ and others where $\eta \propto \sqrt{T}$. Overall, this work offers valuable insights into the optimal scaling of learning rate and batch size as data size increases.

**Strengths:**

In modern LM training, scaling both model size and data size is essential. Understanding how optimal hyperparameters scale with these factors is crucial. While muP provides insights into how hyperparameters scale with model size, less attention has been given to how they scale with data size. This work contributes meaningfully in this direction. Through a set of experiments, the paper offers intriguing insights into how optimal learning rate and batch size scale with increasing data size.

**Weaknesses:**

While this paper addresses an important problem and provides interesting insights, certain clarifications would help strengthen the presentation. Below are some suggestions for improvement:

- **Line 107:** It is mentioned that the "crucial batch size" is defined as the region where $\eta \propto \sqrt{B}$ breaks, and that this comes from Shallue et al. (2019). Could you please point to the specific section in Shallue et al. (2019) where this is discussed?
- **Line 122:** The notation $\epsilon = 1e^{-8}$ seems more like Python syntax. It would be clearer to revise this to $10^{-8}$.
- **Lines 188 and 294:** These lines refer to the "critical batch size" (as a function of data size) as the batch size that leads to the best validation error for a given data size, if I understand it correctly. This definition appears different from the one given in Line 107. Could you please clarify this discrepancy?
- **Lines 265-266:** The paper claims that $B^* \propto \sqrt{T}$. However, in Figure 2a, $B^*$ is either $2^{18}$ or $2^{20}$. How is this square root rule derived with only two data points?
- **Lines 294-296:** The claim that $B_{\text{crit}} \propto T$ is also unclear. Can you explain how this scaling is derived?


For all the claimed scaling laws, if possible, providing the observed data points along with using a scientific fitting method to derive the scaling laws would make the claims more convincing.

I think the paper could be more clear if the speculated laws were better explained and justified.

**Questions:**

See above please.

---

> ### Author Response · Authors · 2024-11-26
>
> Dear Reviewer C7LK,
>
> Thank you for your valuable suggestions, please find our detailed comments to the raised points below:
>
> > Line 107: It is mentioned that the "crucial batch size" is defined as the region where $\eta \propto \sqrt{B}$ breaks, and that this comes from Shallue et al. (2019). Could you please point to the specific section in Shallue et al. (2019) where this is discussed?
>
> As also mentioned in the general comment, we clarified our definition of critical batch size and removed this mentioning. However this is certainly true that Shallue et al. (2019) don’t define it in this way explicitly, but rather present a detailed overview of the diversity of its definition in the literature.
>
> > Line 122: The notation $\epsilon = 1e^{-8}$ seems more like Python syntax. It would be clearer to revise this to $10^{-8}$.
>
> Corrected the notation as suggested
>
> > Lines 188 and 294: These lines refer to the "critical batch size" (as a function of data size) as the batch size that leads to the best validation error for a given data size, if I understand it correctly. This definition appears different from the one given in Line 107. Could you please clarify this discrepancy?
>
> We removed those lines in the revised version due to the change of the critical batch size extraction procedure and its definition. However, to still clarify this, the critical batch size in these two lines wasn’t in conflict with L107 and was matching it exactly. The confusion might’ve come from the fact that the “peak” value of the curve on Fig. 1 doesn’t necessarily mean the best validation error, but rather just the highest possible optimal learning rate for a given token budget across a range of batch sizes. We included (previously Fig. 2, now) Fig. 3 into the main text exactly to relate the optimal (eta, B) values with the corresponding loss value.
>
> > Lines 265-266: The paper claims that $B^* \propto \sqrt{T}$. However, in Figure 2a, $B^*$ is either $2^{18}$ or $2^{20}$. How is this square root rule derived with only two data points?
>
> > Lines 294-296: The claim that $B_{\text{crit}} \propto T$ is also unclear. Can you explain how this scaling is derived?
>
> Addressed in the general comment.
>
> > For all the claimed scaling laws, if possible, providing the observed data points along with using a scientific fitting method to derive the scaling laws would make the claims more convincing.
>
> As mentioned in the general comment, we improved our interpretation of data from the statistical analysis side and clarified the procedure in the main text accordingly. Furthermore, as we introduced additional fitting procedures to extract parameters of interest, we describe the methodology in detail in Appendix A5.

---

> ### Comment · Reviewer_C7LK · 2024-11-28
> **Thank you for your response**
>
> Thank you for your reply. After reading the author’s rebuttal and other reviewers' comments, I decided to maintain my initial rating. Besides other issues, it seems most of the reviewers (if not all) find that the presentation of this paper not satisfactory,  that even affects understanding of the key points of this work (e.g., definitions of critical batch size cause a lot of confusions).
>
> I believe an adequate presentation is necessary for an accurate evaluation of the contribution of a scientific work. I encourage the authors to carefully polish the paper and resubmit it for another round of peer review.

---

> > ### Author Response · Authors · 2024-11-29
> > **Thank you for your feedback**
> >
> > Thank you once again for your revision and valuable feedback.
> >
> > In our rebuttal, we aimed to address all the weaknesses and concerns you highlighted, including clarifying the definitions of key concepts like the critical batch size, which in the current paper version is unambiguously defined according to Eq. 3.1. We also largely expanded our analysis both from the theoretical understanding and explanatory description in order to make it more comprehensive. We hope these updates have improved the clarity and depth of our work.
> >
> > We understand that clear presentation is essential for accurately conveying our contributions, and we made significant efforts to improve these aspects based on your comments. We would greatly appreciate it if you could let us know whether you found our updates helpful or if there are concrete areas that still need improvement? It would be highly valuable for us if you could specify what aspects you feel are still missing or need further clarification.
> >
> > As we would like to enhance the presentation of the paper, your detailed opinion would greatly help us to refine it further.

---

### Official Review · Reviewer_hnCM · 2024-11-05

**Soundness:** 3
**Presentation:** 3
**Contribution:** 2
**Rating:** 6
**Confidence:** 3

**Summary:**

This paper studies the optimal scaling of LLMs in terms of the training hyperparameters, including learning rate $\eta$ and batch size $B$, and their optimal choices with respect to different data sizes $T$. In particular, they show that for different batch size $B$, the optimal $\eta$ has entirely different scaling with $T$. Furthermore, this paper also shows that the optimal batch size is positively correlated with $T$. Then, this paper also shows that the sensitivity with respect to $\eta$ is decreasing when using larger data size $T$. The results can help build a unified picture of the joint hyperparameter tunning of learning rate and batch size.

**Strengths:**

This paper develops extensive experiments on training LLMs with different choices of learning rates, batch sizes, model sizes, and data sizes. Then based on the experimental results, this paper identifies some patterns regarding the optimal hyperparameter configurations. In particular, this paper shows that

* the optimal learning rate has a different evolution in the limit of the data sizes when choosing different batch sizes.
* the optimal batch size is approximately proportion to $\sqrt{T}$ when choosing optimal $\eta$.
* the critical batch size is approximately proportion to $T$ that will affect the optimal choice of learning rate.
* The learning sensitivity can be mitigated when using larger $T$.

Overall these findings provide unified results for guiding the hyperparameter choices when having different token budgets.

**Weaknesses:**

The major weakness of this paper is that it seems to be an experimental report that summarizes the key patterns observed from the experiments, while no further explanation and deeper analysis are conducted. Although there are extensive experimental results and the findings seem to be reliable, it remains unclear whether the findings can be generalized to other settings. I suggest the authors include some understanding of the empirical results, especially about why the learning rate can be much larger when  $T$ is large and we are using a large batch size.

Besides, I am not sure whether only considering $\mu P$ models is sufficient to understand the model with different sizes. It would be good to have more experiments and discussions on the models beyond the $\mu P$ ones.

The definition of $B_{crit}$ is a bit confusing. This may be a kind of clarity issue. The authors begin to mention $B_{crit}$ in the abstract and many places in the introduction. However, the reader cannot quickly get what's the formal definition of $B_{crit}$.

More details about the definition of the effective learning rate for Adam (line 428) should be provided.

**Questions:**

See the weakness section.

---

> ### Author Response · Authors · 2024-11-26
>
> Dear Reviewer hnCM,
>
> Thank you for your valuable comments. We address the major weakness and the issue of the critical batch size definition in the general response, and provide comments to the other raised points below:
>
> > Besides, I am not sure whether only considering $\mu P$  models is sufficient to understand the model with different sizes. It would be good to have more experiments and discussions on the models beyond the $\mu P$  ones.
>
> The main reason for our use of $\mu P$  was to save on compute, because we would be able to transfer results of optimal LR due to the established theory. We agree that pursuing other parametrizations, such as proposed by https://arxiv.org/abs/2407.05872, is a fruitful endeavor and have added this as a proposition for future work. It is also important to mention that we formulated the parametrization so that all models for which $d_\mathrm{model} = d_\mathrm{model}^\mathrm{base}$ are simultaneously in both $\mu P$ and standard parametrization (SP). While we are unable to supply all experimental results in SP due to budget constraints, at least parts of our experiments are thus applicable to models in SP as well.
>
> > More details about the definition of the effective learning rate for Adam (line 428) should be provided.
>
> As we updated the analysis of the optimal learning rate scaling, we removed the previously made noise scale conjecture and the corresponding Eq. 1.

---

> > ### Comment · Reviewer_hnCM · 2024-11-29
> >
> > Thanks for your clarification. It will be helpful for gaining a better understanding of the critical batch size. However, most of the findings are empirical without theoretical interpretations, which is one of the drawbacks of this paper. I will maintain my current evaluation.

---

### Official Review · Reviewer_FvHf · 2024-11-07

**Soundness:** 2
**Presentation:** 2
**Contribution:** 3
**Rating:** 5
**Confidence:** 3

**Summary:**

The authors investigate optimal learning rate, optimal batch size and critical batch size as the number of training training tokens increases.  The authors also investigate the effects of changing mode size within $\mu$P and find that optimal learning rate and batch size do not vary with model size.  The authors also investigate the sensitivity of the loss to learning rate as data and mode size change finding decreased sensitivity as data increases, but similar relationships within $\mu$P scaling.

**Strengths:**

Understanding how optimal learning rate and batch size scale as data scales is a very important and potentially impactful direction.  Understanding the diminishing returns of batch size significantly impact practitioner decisions regarding data parallelism.  Implications here can help go beyond compute optimal scaling paradigm that couples data and model size. In addition, strong understanding of relationships between hyperparameters can help reduce the need for expensive hyperparameter sweeps.

Aligning with existing theoretical models and providing the conjecture of eq. 1 can be useful modes of thinking to eventually provide a deeper understanding of these phenomena.

The scaling of batch size in data presented in Fig 2b and the sensitivity analysis of Fig 3 are both intriguing and useful for the community empirically.

**Weaknesses:**

Overall, the writing needs work. The empirics of this paper are not comprehensive enough to make such a strong claim on training regimes as 3 different growth regimes are being implied from 6 batch sizes and 8 token budgets.  We essentially have 6 trajectories from 8 points and the scaling does not even start immediately because of the "lost" epochs (mainly looking at Fig 1a here).  Given this, the origin of these explicit scaling rules need to be justified by existing work or theory in my opinion.  Content from A.2 should be much more prominent in the main paper.  In addition, the intro feels misleading; the contributions section makes this sound like a theoretical contribution and the specific growth proposed isn't justified by empirics alone.

Fig 2: a 2^2 times budget increase in budget increase of 2^5 is oberved from the sequence with only one step increase?  Extrapolating a general rule from one bump in $B_{\text{crit}}$ is not convincing. In my opinion, it is valuable to look into this, but it is a detrimental to quantitative claims from such little evidence.

**Questions:**

Fig 1a pattern in batch size is not monotone, what's going on here?

Overall, my opinion of the paper is mixed but I lean towards acceptance.  I think much of the empirics are valuable to the community, but the paper is dragged down by overselling some weak empirics.  While I think it is completely reasonable that we don't have comprehensive answers to these questions (presumably from an onerous compute requirement), I'd like to see the claims tempered to mach the strength of the empirics.  I would recommend suggesting further work to investigate what looks like it could be a trend and provide some intuition on why the $T^{-0.5}$, constant and $ T^{0.5}$ regimes are a good conjecture, but hold off on a more forceful claim.

The content of A.2 and I think would be useful to integrate more directly in the main text.

A more rigorous definition of critical batch size (line 107) would be useful.

---

> ### Comment · Reviewer_FvHf · 2024-11-25
> **Reduced score**
>
> After some thought, I have reduced my score.  I was very mixed on this paper, and without addressing concerns I am more comfortable rejecting the paper.  Since the discussion period is still not over, it is possible that this can be changed.

---

> ### Author Response · Authors · 2024-11-26
>
> Dear Reviewer FvHf,
>
> Thank you for the critical review. We address the major weaknesses in the general comment, and provide our comments on the other points below:
>
> > The content of A.2 and I think would be useful to integrate more directly in the main text.
>
> We absolutely agree that this is a vital part to understanding the paper. Due to space constraints posed due to the inclusion of additional figures, we kept A.2 but reworded the main text in the Terminology section 2.1.
>
> > Fig 1a pattern in batch size is not monotone, what’s going on here?
>
> When a large batch size is used with a low token budget, much fewer optimization steps are taken. Thus, the difference in network initialization has a much larger effect on the experiment, leading to higher variance and uncertainty for such points. We hope that the inclusion of additional experiments (4 additional runs per point) and the variance shading helps to make the uncertainty more visible.
>
> > Overall, my opinion of the paper is mixed but I lean towards  acceptance. I think much of the empirics are valuable to the community, but the paper is dragged down by overselling some weak empirics. While I think it is completely reasonable that we don’t have comprehensive answers to these questions (presumably from an onerous compute requirement), I’d like to see the claims tempered to match the strength of the empirics. I would recommend suggesting further work to investigate what looks like it could be a trend and provide some intuition on why the $T^{-0.5}$, constant and $T^{0.5}$ regimes are a good conjecture, but hold off on a more forceful claim.
>
> We thank you for this perspective, which matches our own view on our
> contribution. We hope to have addressed your concerns and matched your
> propositions with our revision.

---

### Official Review · Reviewer_JaN7 · 2024-11-08

**Soundness:** 2
**Presentation:** 3
**Contribution:** 2
**Rating:** 5
**Confidence:** 3

**Summary:**

This paper studies how to set the learning rate $\eta$ and batch size $B$ when the total number of steps $T$ increases. A series of conclusions are drawn from the experiments on MPT of scale 32M, 101M and 354M:
1. A scaling law of the optimal learning rate given $B$. It depends on whether $B$ is smaller than a threshold $B_{crit}$.
2. A scaling law of the optimal batch size, assuming the learning rate is tuned to the optimal.
3. $B_{crit} \propto T$.

**Strengths:**

1. This paper studies an important problem. Hyperparameter tuning is very costly in LLM pretraining. The optimal choices of these hyperparameters are particularly unclear when the number of steps $T$ is a variable.
2. Experiment results are well-organized.
3. A series of scaling laws have been derived. These laws could be useful for tuning the learning rate and batch size.

**Weaknesses:**

My major concern is whether the experiment results can indeed support their scaling laws.
1. It is hard to see the claimed scaling of optimal LR from Figure 1. The curves in Figure 1 are very noisy, and the scaling formulas match the experiment results only for a few choices of $T$. It is thus unclear whether the scaling formulas here describe a real trend or just some noise.
2. It is even harder to see the claimed scaling of optimal batch size from Figure 2. In fact, the optimal batch size only changes once in the figure, which makes me wonder if any scaling concluded from just a single change of optimal batch size could be an overclaim.

Minor weaknesses:
1. It would be good to do some sanity checks on downstream tasks: Does the optimal LR for reducing the pretraining loss also lead to good performance on downstream tasks?
2. Experiments are conducted on only one dataset, C4.

**Questions:**

I wonder if the authors could provide stronger empirical evidence to support the scaling of optimal LR and batch size proposed in the paper.

---

> ### Author Response · Authors · 2024-11-26
>
> Dear Reviewer JaN7,
>
> Thank you for your helpful comments. We address the major concern in the general response, please find our reply to the other points below:
>
> > It would be good to do some sanity checks on downstream tasks: Does the optimal LR for reducing the pretraining loss also lead to good performance on downstream tasks?
>
> The reason that we had not included downstream performance in the first place is that recent works (listed below) have found a correlation between the pre-training loss and downstream performance. However, since this is still an active area of discussion, we agree that an additional sanity check would have been worthwhile. Due to time constraints, we were unable to supply these before the review deadline.
> * Are Emergent Abilities of LLMs a Mirage? https://arxiv.org/abs/2304.15004
> * Compression Represents Intelligence Linearly https://arxiv.org/abs/2404.09937
>
> > Experiments are conducted on only one dataset, C4.
>
> We agree that our results would be more empirically significant and generalizable if we had trained on other data sets. Due to the scope of our experiments and the required computational budget, it would have been computationally infeasible to run the same experiments on other data sets. We do acknowledge that a sanity test on another data set would have been worthwhile. However, C4 is a highly established dataset for LLM research and allows for both better interpretation as well as reproducibility across the general domain.

---

> > ### Comment · Reviewer_JaN7 · 2024-12-02
> >
> > Thanks for your response. However, after skimming over the revised paper, I still feel that the empirical evidence is insufficient to support the claim, as many data points do not really follow the predicted curve. The issue about Figure 2 (now Figure 3), which I mentioned in the review, is still there. I have to keep my score 5 for this reason.
> >
> > Nevertheless, studying the scaling laws and hyperparameter tuning for LLMs is an important direction. I would encourage the authors to keep exploring and submit the paper to the next venue.

---

### Author Response · Authors · 2024-11-26
**General comment**

Dear Reviewers,

We thank you for valuable and critical reviews, and apologise for the additional time required to approach the raised comments in a systematic way. In the general response, we would like to address the points which are common to all of the reviewers, and specific questions are answered separately for each of the reviewers.

The major raised weakness is the gap between empirics and the claimed scaling rules. Also for us, the situation was unsatisfactory. Motivated by the constructive reviews we addressed it in a twofold way: (a) We added further experiments, mainly aiming at reducing the statistical uncertainty of the measured data points. (b) We added thorough analysis based on theory, and now have made major steps towards a comprehensive understanding of the intricate interplay of the different factors. Reflecting this, we have revised our presentation, rewriting the contributions to more accurately reflect our findings.

Specifically, the gap between empirics and claims concerns the (inverse) square root scaling of the optimal learning rate $\eta^*$, the linear increase of the critical batch size $B_\mathrm{crit}$, and the square root increase of the optimal batch size $B^*$ (all with an increase of the token budget $T$). The latter (concluded from old Fig. 2a, now Fig. 3a) is indeed an overclaim and we toned it down to stating objectively that we only observe it increasing with $T$. For the former two, we approach the problem from a deeper perspective:

* We extend our experimental data to reduce statistical noise: Figure 1 now contains 4 additional runs per point for the four highest most sensitive to statistical fluctuations batch sizes. We added variance shading to old Figure 1a (now 1b, we swapped Fig. 1a and 1b in the updated version of the paper) to display this uncertainty.
* We now perform curve fits per token budgets T to data points of old Fig. 1b (now 1a) with the theoretical model of Li et al. (2024) (https://arxiv.org/abs/2405.14578) (Eq. 3.1) and update Fig. 1 with the model fit accordingly.
* We empirically extend the fit model to incorporate the model parameters’ $\eta_\mathrm{crit}$, $B_\mathrm{crit}$ dependence on $T$. By fitting the power law to the measured $\eta_\mathrm{crit}$ and $B_\mathrm{crit}$ data points, we derive their dependence on the token budget $T$ and plot it on the additional Fig. 2. In the formulation of Li et al. (2024),  $B_\mathrm{crit}$ is the critical batch size, and from our fits we observe  $B_\mathrm{crit} \propto T^a$, where $a = 1.0 \pm 0.2$. This is compatible with the claim we made earlier, however now it is more grounded in the statistical method and underlying theory.
* We interpret the fitted model in various regimes of $B$ in relation to  $B_\mathrm{crit}$ and add the corresponding discussion to Sec. 3.2. Our initial claim of square-root / inverse square-root scaling is actually supported by fitted model, but only in the limit of constant $\eta_\mathrm{crit}$, and linear scaling of $b_\mathrm{crit}$ with $T$. We therefore reduce our previous claims in the contribution section to root them into a more objective interpretation of our fit model results.
* We added limitations and future work paragraph in the Discussion section to reflect points raised in the reviews and to provide perspective and directions on the potential improvements.

The last common comment was regarding the critical batch size definition, which we agree was not sufficiently precise before. Since we now ground our interpretation of the data into the theoretical framework of Li et al. (2024), the definition of critical batch size becomes unambiguous, as we update accordingly in Sec. 2.1.

We hope that moderating our claims, inclusion of additional supporting evidence, and deeper analysis of observation suffices to address the gap between empirics and the stated results. In comparison to before the improvement of our analysis based on the theory of Li et al., our paper in general was more descriptive than concise, but with the inclusion of the theory, we judge our own findings now as consistent and very encouraging.

---

### Meta-Review · Area_Chair_6CoQ · 2024-12-19

**Metareview:**

The paper investigates the scaling behavior of optimal learning rate and batch size in the infinite data limit, analyzing their dependence on the pretraining token budget and critical batch size. While the reviewers appreciated the relevance of the topic and the authors' effort to address practical challenges in scaling large language models, significant weaknesses were noted. Key concerns included the lack of theoretical rigor and unclear explanations of empirical findings. Despite clarifications in the rebuttal, these issues remained unresolved. I recommend rejection at this time.

**Additional Comments On Reviewer Discussion:**

During the discussion, reviewers primarily raised concerns about the clarity of the analysis, the presentation of results, and how well the conclusions align with empirical observations. Specific questions were asked regarding the behavior of batch size scaling, assumptions about infinite data limits, and practical applicability. While the authors provided additional clarifications and responses, some doubts remained regarding the precision of the findings and the completeness of the analysis. Weighing these discussions and responses, I concluded that the paper does not yet meet the standard for acceptance.

---

### Decision · Program_Chairs · 2025-01-22

Reject